# Harms of introduced large herbivores outweigh benefits to native biodiversity

Zoé Bescond--Michel [1,2], Sven Bacher [1,3] & Giovanni Vimercati [1,3] ✉

Introduced species significantly impact native biodiversity worldwide, with extensive research on harms but relatively less focus on benefits. Here, using the IUCN Environmental Impact Classification for Alien Taxa (EICAT) and EICAT+ frameworks, we assess 2021 negative and positive impacts of introduced large mammalian herbivores globally. We show that negative impacts are more common, and of higher magnitude than positive impacts, i.e. affect populations, not only the performance of individuals. Native species on islands and at higher trophic level experience greater impacts. Reported impact magnitudes decline over time only for positive impacts. Most positive impacts are caused indirectly through changes in species interactions and ecosystem properties, often following negative impacts on native plants through herbivory and disturbance. We therefore advise caution regarding the intentional introduction of large mammalian herbivores for conservation purposes (rewilding, assisted colonization) without rigorous assessment of their impacts on native communities.

Human-mediated introductions of species outside their native range, so-called alien species, have encompassed all taxonomic groups and geographic regions[1]. A subset of alien species has inflicted substantial harm on native biodiversity, exacerbating its decline alongside other major anthropogenic drivers[1–4]; these are commonly referred to as invasive alien species[5]. Not all alien species harm native biodiversity, and the magnitude of their impacts varies across different alien populations[6]. However, such context dependence of impacts remains poorly understood[7,8], and for most taxonomic groups it is unclear which species are the most harmful and which biological traits and ecological factors determine impact severity. For instance, while islands are generally more susceptible to invasion[9] and more vulnerable to anthropogenic pressures than continents[10,11], it is unclear whether alien species' impacts on insular biodiversity are consistently more pronounced than those on the mainland. Studies have shown that global extinction risk posed by invasive predators on native species is higher on islands[12,13], although it is yet to be determined if the same pattern holds for other negative impact types and magnitudes (reduction in performance of individuals, reduction in population size, local extinction[14]). Similarly, theory predicts that higher trophic levels are more vulnerable to environmental alterations[15–17], but few studies

have investigated if invasive alien species cause stronger negative impacts to native species positioned high in the food chain[18,19].

Native species can also benefit from the introduction of alien species[20,21]. Yet, positive impacts are less often documented than their negative counterparts[20]. While most scientists acknowledge the existence of positive impacts[22], there is a controversy over whether they are overlooked or their extent has been over- or underemphasized in comparison with negative impacts[23–26]. As far as we know, there has been no quantitative, systematically collected and taxonomically controlled study in support of any of these claims, so that the alleged bias of focusing on negative impacts or overstating their magnitude has never been rigorously tested. Additionally, no in-depth investigation on the factors that determine the positive impacts of alien species on native biodiversity has been conducted. A persistent challenge in this regard has been the lack of a transparent and comprehensive framework for measuring and evaluating negative and positive impacts, as well as for effectively comparing their frequencies and magnitudes[21,27–29].

Here, we employ the International Union for Conservation of Nature's (IUCN) Environmental Impact Classification for Alien Taxa (EICAT) framework[14,30] and the recently developed EICAT+

[1]Department of Biology, University of Fribourg, Fribourg, Switzerland. [2]University of Bordeaux, Bordeaux, France. [3]These authors contributed equally: Sven Bacher, Giovanni Vimercati. ✉e-mail: gvimercati@outlook.com

framework[31] to systematically assess negative and positive impacts of introduced large mammalian herbivores (LMH) (LMH; Cetartiodactyla, Perissodactyla, Proboscidea) on native biodiversity on a global scale. While many LMH face dramatic population declines and range contractions because of global change[32], there is a growing call to introduce them outside their native range for conservation purposes, such as trophic rewilding, assisted colonization and climate change mitigation[33–35]. The EICAT(+) frameworks consider native biodiversity as the entity of conservation concern[36] and classify alien species' impacts by their direction, i.e. distinguishing whether they pose harms or offer benefits to local populations of native species. These frameworks also classify impact magnitude into Minimal, Minor, Moderate, Major and Massive levels (see Methods). For this study, assessed impact magnitudes have been further categorized as "weak" or "strong" based on whether they involved individual-level (Minor or lower) or population-level (Moderate or higher) changes to native species. This dichotomous variable is referred to as "impact magnitude" hereafter.

Additionally, the mechanisms through which these impacts were caused have been classified as "direct" or "indirect" (mechanism type). Finally, each impact has been attributed a confidence (low, medium or high) to express the uncertainty associated with the accuracy of the assigned impact magnitude. The combined use of EICAT(+) enables us to conduct a standardized, comparable and taxonomically controlled bidirectional impact assessment needed to address the above controversy. Furthermore, we used the assessed impact data to investigate to what extent insularity and trophic position shape the magnitude of both negative and positive impacts experienced by native species.

Under the assumption that the introduction of species outside their native range mostly disrupts established eco-evolutionary dynamics, we hypothesize that (1) negative impacts caused by introduced LMH on native species occur at a higher frequency and with greater impact magnitude compared to positive impacts. We name this hypothesis the "Harm Dominance Hypothesis". We also hypothesize that (2) both negative and positive impacts of introduced LMH are greater in magnitude on islands and on native species positioned higher in the trophic chain. The hypothesis regarding the influence of insularity and trophic position in amplifying negative impacts stems from circumstantial evidence from previous research[19,37]. Conversely, the hypothesis of greater positive impacts on islands and higher trophic level stems from the rarely tested assumptions that introduced species can restore functions of extinct insular species[34,38], or serve as an important novel food resource for native consumers positioned directly above in the trophic chain[39,40]. Finally, we hypothesize that (3) due to their salience, negative and positive impacts of higher magnitude (strong impacts), such as local extinctions, have been identified first, and thus the reported impact magnitudes across studies would decline over time.

Here, we show that negative impacts of LMH are more common, and of higher magnitude, than positive impacts, while both are greater on islands and at higher trophic levels. We also observe that reported impact magnitudes decline over time only for positive impacts. We conclude that caution is necessary when considering the intentional introduction of LMH for conservation purposes, such as rewilding or assisted colonization, without a rigorous evaluation of their multifaceted impacts on native communities.

## Results

### Frequency of negative and positive impacts

We found 303 reports describing 1616 negative and 405 positive impacts for native species that could be classified under EICAT or EICAT + , from 29 of the 66 listed alien LMH species. Negative and positive impacts were caused by 28 and 21 LMH species, respectively (Fig. 1, Supplementary Data 1). About two thirds of alien LMH species (20 out of 29) caused simultaneously both negative and positive impacts, although for species having bidirectional impacts, we detected 3.7 times more negative than positive impact observations overall (1489 vs. 399, Fig. 1). When comparing these LMH species individually, the trend remained largely consistent (paired sign test: $n = 20$, $p < 0.001$), with records of negative impacts (mean = 74.5 ± 64.6 SD) outnumbering positive impacts (mean = 20 ± 17.3 SD) in all species except two (Bos taurus and Boselaphus tragocamelus, Fig. 1).

Species having exclusively negative impacts in their alien ranges were the Aoudad (Ammotragus lervia), the American bison (Bison bison), the Wapiti (Cervus canadensis), the Asian elephant (Elephas maximus), the Guanaco (Lama guanicoe), the Gemsbok (Oryx gazella), the Mouflon (Ovis orientalis), and the Javan deer (Rusa timorensis). Only one species, the Indian hog deer (Axis porcinus), had exclusively positive impacts, Fig. 1.

A great majority ($n = 27$; 93%) of alien LMH species for which impacts are reported caused strong impacts (positive or negative). Almost all the 28 species with negative impacts caused strong impacts (93%, $n = 26$). By contrast, among the 21 species causing positive impacts, only 71% ($n = 15$) caused strong impacts (Fig. 1). Observations of negative impacts outnumbered those of positive impacts across all levels of magnitude (Fig. 1, Supplementary Fig. 1) and confidence (Supplementary Fig. 2).

The predominant impact magnitude observed was Moderate (MO and MO + , Supplementary Fig. 1., Supplementary Data 1), with native population decline documented at 52% (840/1616) and native population increase at 45% (184/407). Across the five impact magnitude levels, confidence was mostly categorized as low and medium, while a high confidence was less frequently assigned (Supplementary Fig. 2). This trend was consistent for both negative and positive impacts, except for cases where alien LMH increased the size of native populations (MO + ). These cases were assigned with significantly higher confidence compared to instances of native population decreases (MO) (z-test, $z = -4.1$, $p < 0.001$, Supplementary Fig. 2).

Overall, alien LMH caused negative impacts mostly through direct mechanisms (direct = 78%, denoted by black labels in Fig. 2), while the opposite trend was observed for positive impacts (indirect = 85%, denoted by green labels in Fig. 2, Supplementary Table 1). The most frequently recorded mechanism for negative impacts was direct "grazing, herbivory, or browsing" ($n = 982$ impacts), followed by indirect "chemical, physical, or structural impacts on ecosystems" ($n = 314$ impacts), and direct "bio-fouling or other direct physical disturbances" ($n = 296$ impacts). Conversely, "indirect impact through interactions with other taxa" ($n = 275$ impacts) was the predominant mechanism through which positive impacts were caused, followed by indirect "chemical/physical/structural impact on the ecosystem" ($n = 74$ impacts).

Negative impacts of alien LMH were more frequently documented on islands (68% of all reports of negative impacts), whereas positive impacts showed a more even distribution (51% from islands, 49% from mainland) (Fig. 3B). Impacts from alien LMH affected four trophic levels: decomposers, producers, primary consumers, and secondary consumers. The trophic level most frequently impacted, both negatively (74%) and positively (59%), was producers (Fig. 3D).

### Predictors of impact magnitude

Among the 511 models obtained, each representing different combinations of variables and their interactions with impact direction, 12 exhibited a ΔAICc <6 in relation to the best model (Table 1). Thus, we performed model averaging and estimated the relative importance (sum of Akaike weights) of each factor and interaction within the selected set of models (Table 2).

After averaging models across the 13 best-fitting candidates, predictors (both factors and interactions) demonstrating sufficient explanatory power (relative importance > 0.5, Table 2) aligned with those featured in the model characterized by the lowest AICc (best-

fitting model, Table 1). We therefore selected the best-fitting model (Tables 1, 3) as the most supported model for further analyses, pairwise comparisons and data visualizations.

Globally, alien LMH species exhibited a higher probability to cause strong negative impacts than positive impacts (Table 3, Fig. 3A,

Supplementary Table 2). Moreover, both negative and positive impacts were stronger in insular locations compared to mainland locations (Table 3, Fig. 4A, Supplementary Table 2). Over the years, we detected a non-significant overall decrease in the probability of causing strong impacts, with a steeper decline in the magnitude of positive

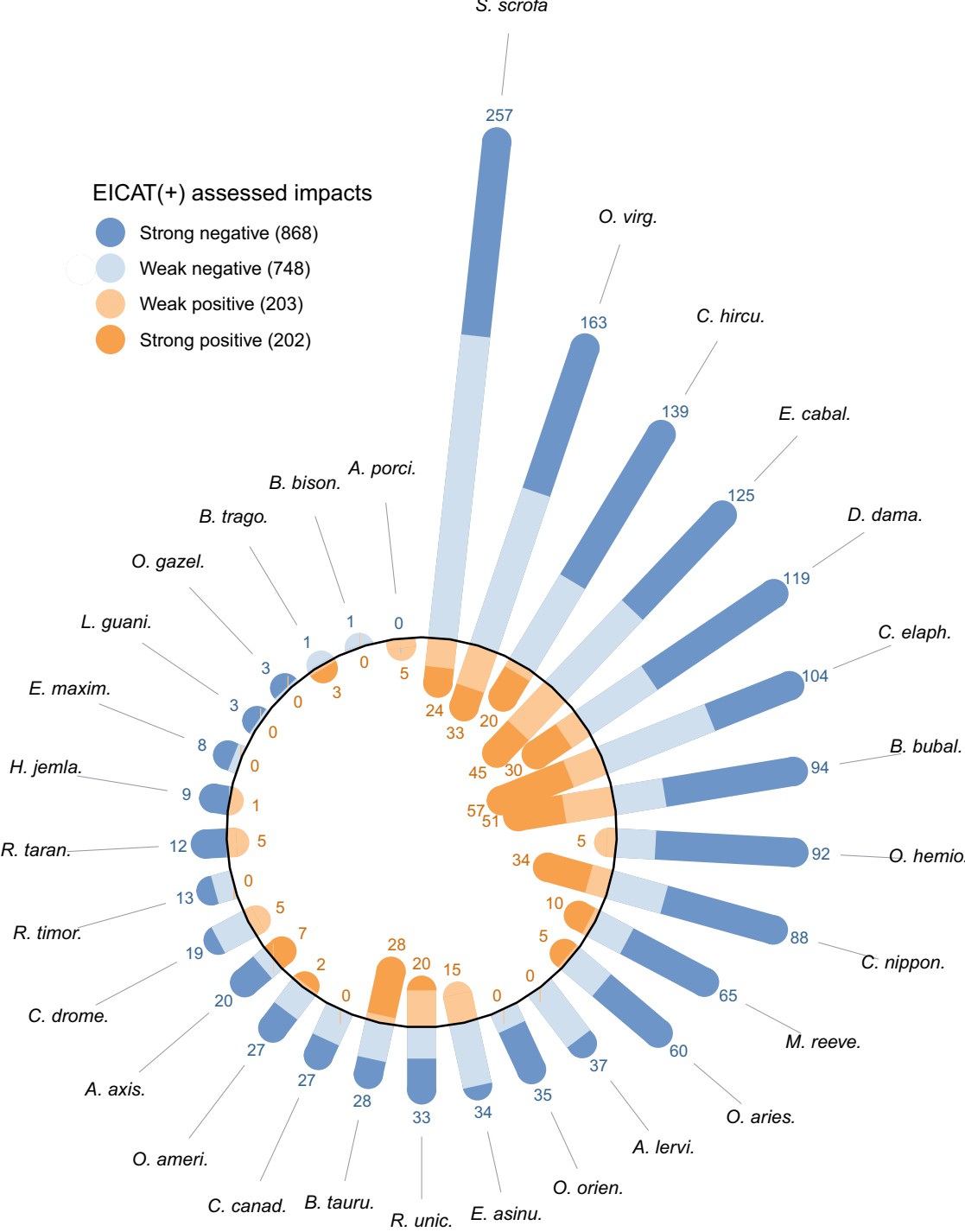

**Fig. 1 | Number of assessed impacts for each introduced LMH species, categorized by impact direction and magnitude.** Number of negative (blue) and positive (orange) impact observations (*N* = 2021) for 29 introduced LMH species assessed with EICAT(+). Pale and dark shades represent weak and strong impacts, respectively. Numbers represent the sample size for positive and negative impact observations for each species. Abbreviations stand for the following species names: *S. scrofa* = *Sus scrofa*; *O. virg.* = *Odocoileus virginianus*; *C. hircu.* = *Capra hircus*; *E. cabal.* = *Equus caballus*; *D. dama.* = *Dama dama*; *C. elaph.* = *Cervus elaphus*; *B. bubal.*

= *Bubalus bubalis*; *O. hemio.* = *Odocoileus hemionus*; *C. nippon.* = *Cervus nippon*; *M. reeve.* = *Muntiacus reevesi*; *O. aries.* = *Ovis aries*; *A. lervi.* = *Ammotragus lervia*; *O. orien.* = *Ovis orientalis*; *E. asinu.* = *Equus asinus*; *R. unic.* = *Rusa unicolor*; *B. tauru.* = *Bos taurus*; *C. canad.* = *Cervus canadensis*; *O. ameri.* = *Oreamnos americanus*; *A. axis.* = *Axis axis*; *C. drome.* = *Camelus dromedarius*; *R. timor.* = *Rusa timorensis*; *R. taran.* = *Rangifer tarandus*; *H. jemla.* = *Hemitragus jemlahicus*; *E. maxim.* = *Elephas maximus*; *L. guani.* = *Lama guanicoe*; *O. gazel.* = *Oryx gazella*; *B. trago.* = *Boselaphus tragocamelus*; *B. bison.* = *Bison bison*; *A. porci.* = *Axis porcinus*.

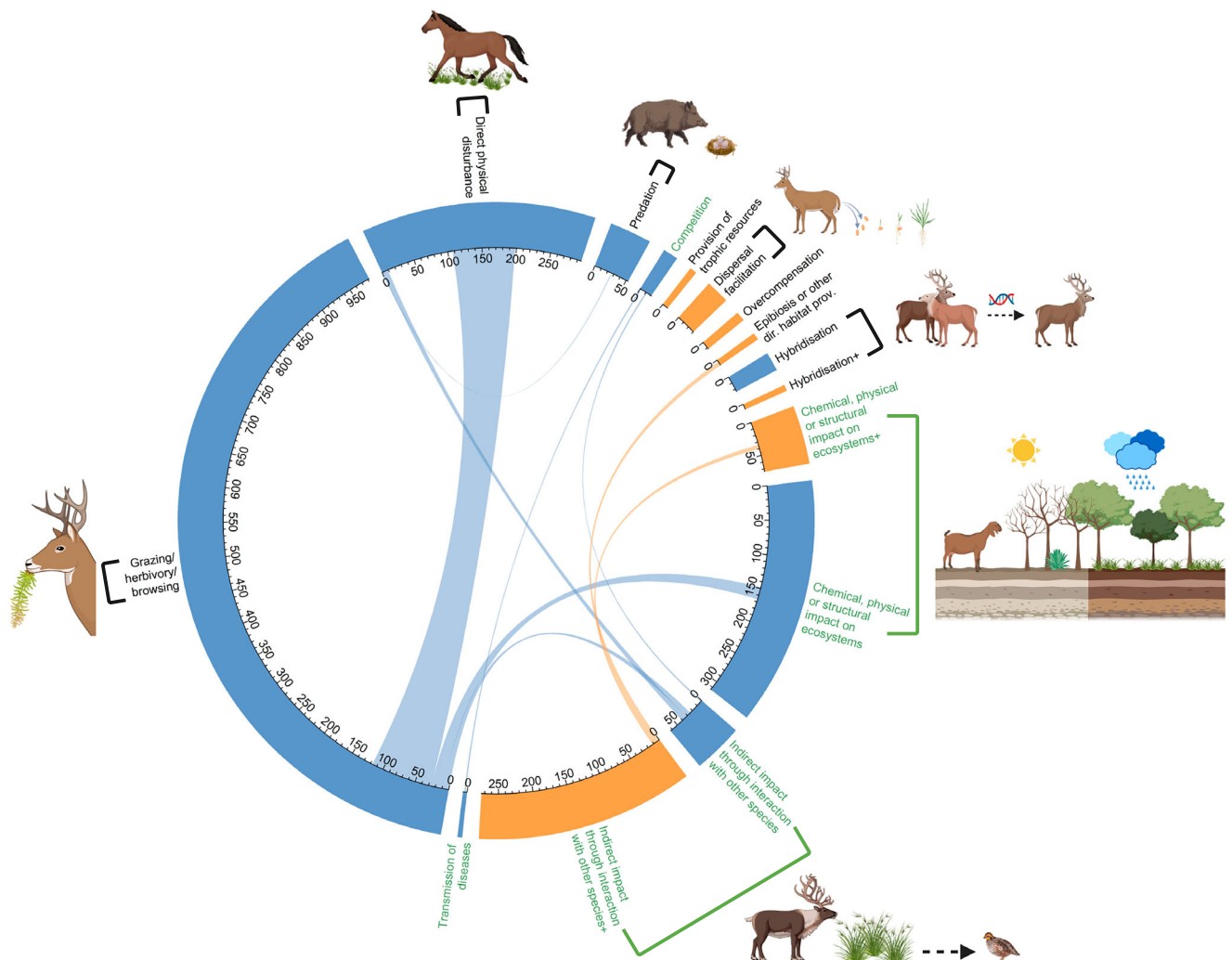

**Fig. 2 | Number of assessed impacts for each impact mechanism across introduced LMH species.** Number of negative (blue) and positive (orange) impact observations ($N = 2021$) for each mechanism, and assessed with EICAT(+). Black and green labels represent direct and indirect mechanisms, respectively. Arrows represent impact observations in which at least two mechanisms were jointly assigned to the same observed impacts. Mechanisms through which introduced LMH cause a substantial number (> 20) of negative and positive impacts on native biodiversity are depicted in the figure. Note that through the mechanisms "Hybridization", "Chemical, physical or structural impact on ecosystems" and "Indirect impacts through interaction with other species", alien LMH caused both negative and positive impacts. Created in BioRender. Vimercati, G. (2025) https://BioRender.com/0s3bd0e.

compared to negative impacts (Table 3, Fig. 4B). Regardless of impact direction, alien LMH species were more likely to cause strong impacts on insular than on mainland locations (Fig. 3B, Supplementary Table 2), through indirect compared to direct impact mechanisms (Fig. 3C, Supplementary Table 2), and on secondary consumers compared to primary consumers (Fig. 3D, Supplementary Table 2). Conversely, the effects on other trophic levels (producers and decomposers) were indistinguishable (Fig. 3D, Supplementary Table 2).

**Confidence in assigning impact magnitude**

The complete model including Confidence, Direction, Year and their 2-way interactions as predictors of impact magnitude strongly outperformed all simpler models ($\Delta$AICc = 6.45 from the second-best model). According to this model (Table 4), strong impacts were assigned with higher confidence than weak impacts, regardless of impact direction (Fig. 5A, Supplementary Table 2), but the rise in confidence with impact magnitude was steeper in positive than in negative impacts (Fig. 5B, Supplementary Table 2). Moreover, the probability of causing strong impacts decreased significantly faster over the years for impacts classified with high and

medium confidence than for those classified with low confidence (Fig. 5C).

## Discussion

The introduction of LMH outside their native range has both harmed and benefited local native biodiversity, but negative consequences have largely surpassed positive outcomes, both in frequency and magnitude. Here, we comprehensively compared the negative and positive impacts of alien species and identified factors determining their magnitude. By leveraging the methodological advances of the EICAT(+) frameworks, we systematically tested hypotheses that were previously only supported anecdotally for negative impacts and never tested for positive impacts. This enabled us to provide a rigorous and detailed examination of how species that have established alien populations impact native biodiversity, demonstrating that the magnitude of both their negative and positive impacts is influenced by common factors such as insularity and trophic position.

The observed negative impacts of alien LMH disproportionately outnumber positive impacts on a global scale. This overall pattern does not arise solely from a few highly impactful taxa but remains consistent when examining species individually (Fig. 1), and when

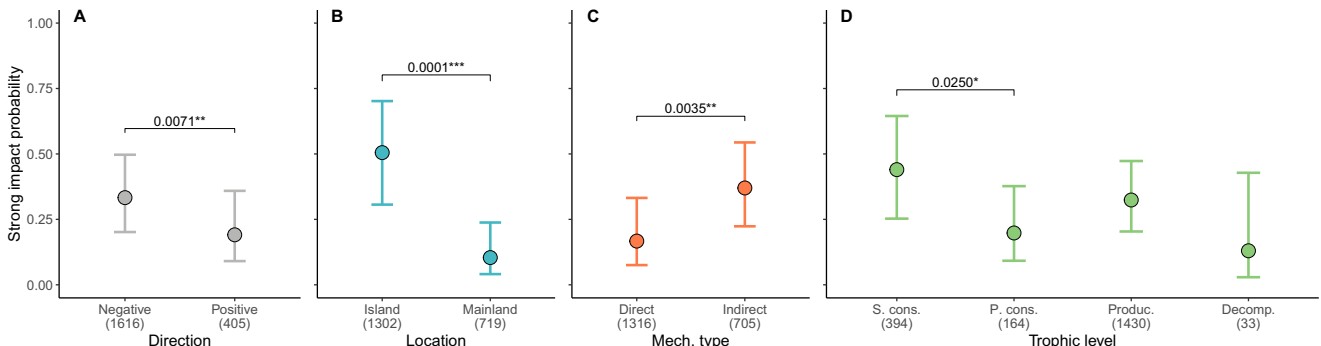

**Fig. 3 | Effects of direction, location, mechanism type, and trophic level on impact magnitude across introduced LMH species. A–D** Estimated probabilities of introduced LMH species causing strong impacts on native biodiversity, based on the most supported generalized linear mixed-effects model (Table 3). Circles represent estimated marginal means across predictor levels, with bars indicating 95% confidence intervals. Post hoc pairwise comparisons were conducted using Tukey's Honest Significant Difference correction for multiple comparisons, with two-sided tests at a 95% confidence level. Significant differences were observed across **A** impact directions (gray), **B** locations (blue), **C** mechanism types (red), and **D** trophic levels (green). Sample sizes for each group are shown in brackets. Horizontal brackets and asterisks denote statistically significant differences: \*\*\**p* < 0.001; \*\**p* < 0.01; \**p* < 0.05. Abbreviations: S. cons. = secondary consumers; P. cons. = primary consumers; Produc. = producers; Decomp. = decomposers. Full p-values from post hoc comparisons are reported in Supplementary Table 2.

focusing on mechanisms through which both negative and positive impacts can be caused (Fig. 2, Supplementary Table 1). Our finding that only 20% of all impacts of alien LMH are positive (405/2021) aligns with findings from other systematic searches for positive and negative impacts. The recent IPBES report classified 15% of documented alien species impacts on nature from all taxonomic groups as positive[20], while a study by Chen and coworkers[41] on alien freshwater megafish found only 3% positive environmental impacts. However, three potential biases in the dataset we compiled–namely, challenges in accessing literature on positive impacts, a historical focus on studying and reporting negative impacts, and a predominance of negative impacts for populations erroneously considered alien despite being introduced within their historically native range–may have contributed to the observed imbalance in the number of negative and positive impact observations. By using search strings that specifically included terms like "positive impact," "beneficial impact," and "benefit," and by cross-checking literature on negative impacts to source additional examples of positive outcomes, we expanded our capacity to capture a broad spectrum of positive impacts that may have been overlooked in previous analyses. However, our search strategy could not overcome the possible inherent bias resulting from the predominant research focus on negative impacts, a tendency that has been hypothesized before[23–26], but has not been supported with evidence. Without knowing if and how much researchers have preferentially chosen study systems in which they expected to find predominantly negative impacts, it is impossible to correct for such bias in analyses on impact numbers. Nonetheless, a very similar outcome could also arise if negative impacts genuinely outnumber positive ones, making it difficult to disentangle these two mechanisms. Although a possible bias cannot be conclusively ruled out, several indications suggest it does not undermine the main conclusions of our research. Among the subset of 51 reports (out of 303) that documented both positive and negative impacts for native species–i.e., reports whose study design and methods allowed the authors to capture changes in both directions–negative impacts outnumbered positive impacts (mean per report: 8.9 ± 9.6 SD vs. 6.6 ± 9.4 SD; paired sign test: *p* < 0.001). Moreover, among reports that exclusively documented negative or positive impacts (231 vs. 21), the proportion of reports with a single observation was similar between the two groups (35 vs. 48%). This suggests that although some studies may have exhibited confirmation bias by selectively picking cases to highlight specific impacts on native species[42], rather than conducting less biased investigations across a broader range of species, this bias affected positive and negative impact studies equally.

In addition, temporal reporting of positive impacts was not increasing faster than that of negative impacts. In fact, the number of positive impact observations has remained quite stable over the last decades (Supplementary Fig. 1.), despite the recent popularity of literature emphasizing the necessity to acknowledge positive impacts of alien species for conservation purposes[25,42–48]. Furthermore, the magnitude of reported positive impacts declined faster over the years than that of negative impacts (Fig. 4B) indicating that strong positive impacts may have been incrementally more difficult to identify than negative ones. The above considerations suggest that the greater number and severity of negative impacts are not primarily due to reporting bias but reflect an inherent asymmetry in how alien LMH affect native biodiversity.

In our study, a minority of impact reports (455 out of 2021, Supplementary Data 1) were from areas inside or adjacent to the native continent of the introduced species and thus might have been located in areas formerly inhabited by the species (refugee species concept RSC[49]). The RSC suggests that species ranges of some LMH may have been drastically reduced over time by humans, and extant populations are currently confined to refuge habitat that might not be optimal[50]. In accordance with the RSC, introductions near the currently known native range might indicate locations within the historical native ranges of the introduced species but whose relatively recent disappearance led authors of the impact reports we compiled to mistakenly treat them as alien[51]. It could be hypothesized that the impact magnitude or direction may both vary with their degree of alienness. Therefore, beside our main analysis, we ran an additional analysis with a broader set of predictors, incorporating a binomial predictor designating the introduction location as either "true alien" or "potential reintroduction to former native area" (i.e., on the same continent as the native range, or in cases where the native range, e.g., Northern Africa or Papua New Guinea, borders on an introduction location on a different continent, e.g. Southern Spain, Indonesia in the close vicinity). However, neither this predictor nor its interaction with impact direction improved model fit (relative importance = 0.41 and 0.20, respectively, both well below the threshold of 0.5, included in only 17 and 8 of the 30 selected best-fitting models), and were therefore dropped during the model selection process. This suggests that the uncertainty surrounding the biogeographic status of some introduced populations of LMH has little influence on the observed impact patterns and is therefore unlikely to bias our results.

Finally, a great majority of all alien LMH species studied for their environmental impacts (26 of 29) caused negative population level impacts (Fig. 1), and through disparate mechanisms, such as herbivory,

**Table 1 | Degrees of freedom (df), values of AICc (corrected Akaike Information Criterion), ΔAICc, Akaike weight and evidence ratio (ω1/ω, where ω1 is the Akaike weight of the best-fitting model) of 13 best-fitting generalized linear mixed-effect models (ΔAICc <6) predicting impact magnitude of alien LMH**

| Fixed effects | df | AICc | ΔAICc | weight | Evidence ratio |
|---|---|---|---|---|---|
| D, L, M, TL, Y, D*L, D*Y (best-fitting model) | 12 | 1689.43 | 0 | 0.22 | - |
| D, L, M, TL, Y, D*Y | 11 | 1689.47 | 0.04 | 0.21 | 1.03 |
| D, L, TL, Y, D*L, D*Y, D*TL | 14 | 1690.02 | 0.59 | 0.16 | 1.31 |
| D, L, M, TL, Y, D*L, D*M, D*Y | 13 | 1691.38 | 1.94 | 0.08 | 2.61 |
| D, L, M, TL, Y, D*M, D*Y | 12 | 1691.49 | 2.06 | 0.08 | 2.80 |
| D, L, M, TL, Y, D*L, D*Y, D*TL | 15 | 1691.78 | 2.35 | 0.07 | 3.10 |
| D, L, Y, TL, D*Y, D*TL | 13 | 1692.56 | 3.12 | 0.05 | 4.71 |
| D, L, TL, Y, D*Y | 10 | 1693.31 | 3.88 | 0.03 | 6.61 |
| D, L, M, TL, Y, D*L, D*M, D*Y, D*TL | 16 | 1693.34 | 3.91 | 0.03 | 7. 11 |
| D, L, M, TL, Y, D*Y, D*TL | 14 | 1694.24 | 4.81 | 0.02 | 10.78 |
| D, L, M, Y, D*L, D*Y | 9 | 1694.24 | 4.81 | 0.02 | 11.43 |
| D, L, M, Y, D*Y | 8 | 1694.84 | 5.41 | 0.01 | 15.56 |
| D, L, TL, Y, D*L, D*Y | 11 | 1695.28 | 5.85 | 0.01 | 18.82 |

In the first column, the letter 'D' stands for Direction, 'L' for Location, 'M' for Mechanism types, 'Y' for Year, and 'TL' for Trophic level, while the symbol "*" indicates an interaction between two variables. Note that all models retain "Alien species name" and "Report ID" as random effects.

**Table 2 | The relative importance, indicated by the sum of Akaike weights, of each predictor (variables and interaction between variables) along with the number of models in which they appear, as assessed among the 13 best-fitting generalized linear mixed-effect models (ΔAICc <6) predicting impact magnitude of alien LMH**

| Predictor | D | L | Y | D*Y | TL | M | D*L | D*TL | D*M |
|---|---|---|---|---|---|---|---|---|---|
| Relative importance | 1 | 1 | 1 | 1 | 0.97 | 0.75 | 0.60 | 0.33 | 0.19 |
| # best-fitting models containing the predictor / # best-fitting models | 13 / 13 | 13 /13 | 13 / 13 | 13 / 13 | 11 / 13 | 9 / 13 | 7 / 13 | 5 / 13 | 3 / 13 |

In the column header, the letter 'D' stands for Direction, 'L' for Location, 'M' for Mechanism types, 'Y' for Year, and 'TL' for Trophic level, while the symbol "*" indicates an interaction between two variables.

**Table 3 | Results from the most supported generalized linear mixed-effect model (with Alien species name and Report ID as random effect) testing the overall effects of Direction, Location, Mechanism type, Trophic level and the interaction between Direction and Location, as well as Year, on the probability of an alien LMH species causing a strong impact on native biodiversity**

| Predictor | Chi-square | df | p |
|---|---|---|---|
| Direction | 3.79 | 1 | 0.052 |
| Location | 18.25 | 1 | <0.001*** |
| Mechanism type | 8.52 | 3 | 0.004** |
| Year | 0.73 | 1 | 0.393 |
| Trophic level | 10.73 | 1 | 0.013* |
| Direction * Location | 2.35 | 1 | 0.126 |
| Direction * Year | 13.888 | 1 | <0.001*** |

Asterisks indicate significant differences: ***$p < 0.001$; **$p < 0.01$; *$p < 0.05$. All p-values are based on two-sided tests.

**Table 4 | Results from the most supported generalized linear mixed-effect model (with Alien species name and Report ID as random effects) testing the overall effects of Direction, Confidence, Year and the interactions between them on the probability of an alien LMH species causing a strong impact on native biodiversity**

| Predictor | Chi-square | df | P |
|---|---|---|---|
| Direction | 3.31 | 1 | 0.069 |
| Year | 4.47 | 1 | 0.034* |
| Confidence | 25.07 | 2 | <0.001*** |
| Direction * Year | 14.43 | 1 | <0.001*** |
| Direction * Confidence | 30.45 | 2 | <0.001*** |
| Confidence * Year | 10.51 | 2 | 0.005** |

Asterisks indicate significant differences: ***$p < 0.001$; **$p < 0.01$; *$p < 0.05$. All p-values are based on two-sided tests.

direct physical disturbance, hybridization, interactions with other species, and indirect impacts on ecosystems (Fig. 2, Supplementary Table 1). Almost half of the studied alien LMH species (14), conversely, did not have documented positive impacts at the population level (Fig. 1). Accordingly, positive impacts were overall characterized by lower magnitude than their negative counterparts (Figs. 3A, 4A), thus further supporting the "Harm Dominance Hypothesis". Strong positive impacts were also predominantly caused through indirect mechanisms (195 cases out of 202, Fig. 1), while direct mechanisms (i.e., provision of trophic resources and habitat or overcompensation) rarely led to population level impacts (7). Since strong positive impacts on native biodiversity were caused by alien LMH mostly via indirect impacts through interactions with other species (167), and often the latter were species negatively affected by the same alien LMH, we suggest that negative impacts often precede positive ones. Particularly insightful are cases where the same species exhibited both strong negative and strong positive impacts, with the negative impacts consistently outnumbering the positive (Fig. 1, $p < 0.001$). For instance, the grazing pressure imposed by introduced goats (*Capra hircus*) on native vegetation caused a decline in the abundance of 66 insular plant species, and three instances of extirpation were also reported (Fig. 1).

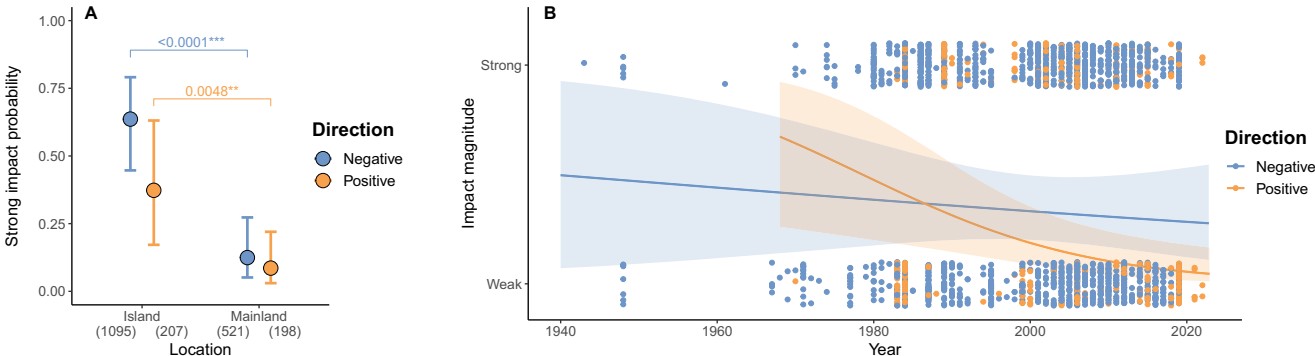

**Fig. 4 | Effects of location and reporting impact year on the magnitude of positive and negative impacts across introduced LMH species. A** Estimated probabilities of introduced LMH species causing strong impacts on native biodiversity, based on the most supported generalized linear mixed-effects model (Table 3). Circles represent estimated marginal means across predictor levels, with bars indicating 95% confidence intervals. Post hoc pairwise comparisons were conducted using Tukey's Honest Significant Difference correction for multiple comparisons, with two-sided tests at a 95% confidence level. Significant differences between island and mainland locations were observed, consistent for both negative (blue) and positive (orange) impacts. Sample sizes for each group are shown in brackets. Horizontal brackets and asterisks denote statistically significant differences: ***$p < 0.001$; **$p < 0.01$; *$p < 0.05$. Full p-values from post hoc comparisons

are reported in Supplementary Table 2. **B** The regression slopes of estimated marginal means across years indicate that the probability of strong impacts changes significantly over time depending on impact direction ($p < 0.001$), based on the most supported generalized linear mixed-effects model (Table 3). Specifically, the probability of causing a strong positive (orange) impact showed a steeper decline over time compared to negative (blue) impacts. Differences in slopes were tested using pairwise comparisons of model-estimated trends, with two-sided tests at a 95% confidence level and no adjustment for multiple comparisons. Shaded areas represent 95% confidence intervals (CIs) around the slope estimates. Note the wide CIs for both slope estimates. Dots represent individual impact observations across years and levels of impact magnitude, with a jitter function applied to reduce overlap.

Conversely, only 13 plants have increased their abundance after the introduction of goats to islands. This positive effect was primarily observed on unpalatable ferns and monocotyledonous species, which benefited from competitive release as the goats preferentially fed on more palatable broadleaved plants[52]. Similar ecosystem changes from woodlands to grasslands (including ferns) were promoted by widespread alien deer species such as *Cervus elaphus*[52,53], *Cervus nippon*[54] and *Muntiacus reevesi*[55]. Under some circumstances alien LMH significantly benefit native plant species that are less abundant in native communities by releasing them from their competitors. Such a positive outcome is achieved at the expense of more competitive native species that are suppressed by the same alien LMH (see mechanism "Interaction with other species" in Fig. 2). Thus, many positive impacts generally do not occur directly, but only indirectly, after other native species suffer. If positive impacts are often due to the prior occurrence of negative impacts – which can conversely occur independently of positive impacts through mechanisms such as herbivory or direct disturbance (Fig. 2) – this could partially explain why the number of negative impacts of alien species is larger overall.

In accordance with our predictions, both negative and positive impacts of alien LMH were larger on islands (Fig. 3). The effect of insularity on impact magnitude is especially evident for negative impacts (Fig. 4A), supporting the hypothesis that insular biodiversity is particularly vulnerable to anthropogenic alterations[10,11]. Previous studies have shown that species on islands were driven towards local or global extinction primarily by predatory effects from a few widely introduced mammals such as rats, mongooses, wild boars, and feral cats and dogs[37,56]. Predation is the most widely cited mechanism for biodiversity decline on islands[57–60]. In our study, most predation events (49 out of 52) were by wild boars (*Sus scrofa*), an omnivorous species that feeds predominantly on plant matter and opportunistically consumes invertebrates, small vertebrates and eggs[61], especially in the introduced range[62]. The other three predation events were by feral goats (*Capra hircus*). Notably, all recorded predatory events that led to strong impacts occurred on islands: wild boars caused population declines in two insular lizards, three seabirds, and one rail species; feral goats caused population declines in three insular seabirds. Although our results confirm the vulnerability of native biodiversity to predation on islands, the vast majority of strong negative impacts by LMH on

islands (670 out of 679) were caused by other mechanisms, including direct physical disturbance (42), chemical, physical, or structural impacts on ecosystems (166), and grazing/herbivory/browsing (375), as well cases where two or more of these mechanisms were jointly assigned (Fig. 2). Since LHM species caused negative impacts through the same mechanisms also on the mainland, our results substantiate the rarely tested assumption that native biodiversity on islands is particularly vulnerable to impacts of alien species, regardless of the mechanisms.

Our findings highlight the unique and sensitive nature of insular ecosystems, where positive impacts of alien species are also higher in magnitude. Alien species can facilitate native biodiversity by restoring functions previously held by recently extinct or extirpated species[47], particularly on islands[34,38]. However, our data on alien LMH do not conclusively support the functional replacement hypothesis. For instance, alien wild boars, feral goats, Reeves' muntjacs, and mule deer have facilitated the dispersal of native plants on islands, but their positive impacts were weak, meaning they did not increase native plant populations. Only a few strong positive impacts on islands were caused through chemical, physical, and structural impacts on ecosystems ($N = 4$), epibiosis or other direct habitat provisions (1), overcompensation (1), and provision of trophic resources (1). Alien species had positive population-level impacts mainly through interactions with other species (113), mostly benefiting plants (98) by reducing the grazing or browsing pressure on their direct native competitors. The higher magnitude of positive impacts on islands might be an indirect consequence of the initial decline caused by alien LMH on insular biodiversity.

We found that native species at higher trophic levels (secondary consumers) were more impacted by alien LMH than those at lower trophic levels. While there is evidence that top trophic levels are more sensitive to environmental change[17], our study conclusively demonstrates this across multiple terrestrial taxa. Previous studies have mainly explored this relationship within single taxa or taxonomic levels, or only in marine communities. For example, terrestrial alien plants have caused various negative impacts on higher trophic levels[18,19,63,64], but it remains unclear if these impacts are larger, equal to, or smaller than those on native producers[65]. Thomsen and coworkers[66] found that alien marine producers and consumers

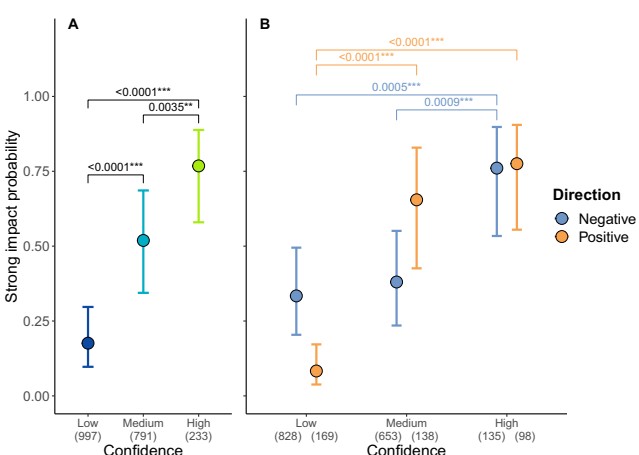

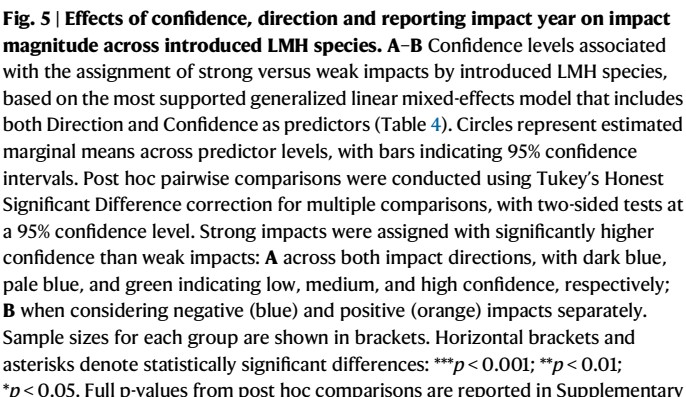

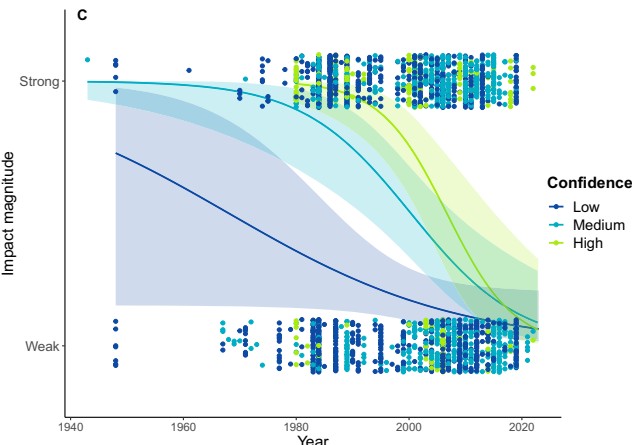

**Fig. 5 | Effects of confidence, direction and reporting impact year on impact magnitude across introduced LMH species. A–B** Confidence levels associated with the assignment of strong versus weak impacts by introduced LMH species, based on the most supported generalized linear mixed-effects model that includes both Direction and Confidence as predictors (Table 4). Circles represent estimated marginal means across predictor levels, with bars indicating 95% confidence intervals. Post hoc pairwise comparisons were conducted using Tukey's Honest Significant Difference correction for multiple comparisons, with two-sided tests at a 95% confidence level. Strong impacts were assigned with significantly higher confidence than weak impacts: **A** across both impact directions, with dark blue, pale blue, and green indicating low, medium, and high confidence, respectively; **B** when considering negative (blue) and positive (orange) impacts separately. Sample sizes for each group are shown in brackets. Horizontal brackets and asterisks denote statistically significant differences: ***$p < 0.001$; **$p < 0.01$; *$p < 0.05$. Full p-values from post hoc comparisons are reported in Supplementary

Table 2. **C** The regression slopes of estimated marginal means across years indicate that the probability of introduced LMH species causing strong impacts on native biodiversity changes significantly over time depending on the level of confidence assigned to the impact, based on the most supported generalized linear mixed-effects model that includes both Direction and Confidence as predictors (Table 4). Specifically, impacts classified with high (green) and medium (pale blue) confidence showed a steeper decline over time compared to those classified with low (dark blue) confidence (High–Low: $p < 0.01$; Medium–Low: $p < 0.05$). No significant difference was detected between high and medium confidence levels ($p = 0.12$). Differences in slopes were tested using pairwise comparisons of model-estimated trends, with two-sided tests at a 95% confidence level and no adjustment for multiple comparisons. Shaded areas represent 95% confidence intervals around the slope estimates. Dots represent individual impact observations across years and confidence levels, with a jitter function applied to reduce overlap.

negatively impact native species within their trophic level rather than higher ones, mainly through competition and other antagonistic interactions. They also found that introduced species can serve as significant novel food resources for native consumers, benefiting species positioned directly above in the trophic chain. Our results suggest that introduced LMH have severe impacts on high-trophic-level species, mostly through indirect interactions or ecosystem changes. In contrast, direct impacts on species at the same or lower trophic levels through antagonistic interactions like competition, predation, or herbivory have lower impact magnitudes. While species high in the food chain might be particularly vulnerable to alien species, our findings stress the need for community-level studies that include complex indirect interactions beyond direct individual species interactions.

Similar considerations may apply to positive impacts. Studies on native predators feeding on alien species[67–70], pollinators utilizing alien plant nectar and pollen[71] and frugivores incorporating alien fruits in their diets[72] found that alien species can benefit species directly above them in the trophic chain by providing trophic resources[39,40,66]. However, among all positive impacts of alien LMH on secondary consumers (114), only nine (8%) were through food provision, with only one having population-level consequences. Instead, alien LMH benefited secondary consumers mostly indirectly through ecosystem changes (53) and interactions with other species (50), leading to strong population-level positive impacts in the majority of cases (60 out of 103). We conclude that species high in the food chain can sometimes benefit from complex trophic cascades or habitat provisioning initiated by alien species introductions, while direct provision of trophic resources plays a minor role in affecting local biodiversity.

We did not find support for our hypothesis that regardless of impact direction, strong impacts are reported first and thus impact magnitude would decline over time. Notably, we found that impact magnitude steeply declines over time for positive impacts (Fig. 4B),

whereas the decline for negative impacts was much shallower and non-significant (Table 3). This may indicate that strong positive impacts, i.e., those concerning population level changes induced by alien LMH on native species, were identified, and therefore reported, first due to their obvious extent. Conversely, positive impacts having weaker magnitude levels, i.e., involving individuals rather than populations, might have been initially less evident and remained undetected for years.

Alternatively, improved analytical methods might have recently revealed that positive impacts often affect native individuals without significant population-level consequences. This latter conjecture might be supported by the finding that the magnitude of positive impacts classified with high and medium confidence decreases more steeply than low-confidence impacts. This indicates that population-level positive impacts assigned with greater certainty become scarcer over time in favor of those assigned with analogous levels of confidence but measured at the individual level. It is also worth noting that while weak impacts also encompass Minimal positive impacts (see Methods), they have been reported more often (64 vs 46) and with higher confidence (high/medium = 52 vs 6%) in the last two decades (2000–2019) than in the previous two decades (1980–1999). We anticipate this trend will continue, as our research identified several instances where positive impacts at the individual level might exist. However, the study design or the use of composite biodiversity indicators (such as species richness, diversity, or evenness) did not allow us to conclusively determine the magnitude of these impacts. For example, future studies will likely elucidate to which extent feral donkeys in the Sonoran Desert, which are preyed upon by native cougars[73] and play a role in shaping dryland ecosystems by increasing water availability[74], benefit native species, but also which other native species might suffer.

Since the majority of LMH species are threatened by extinction in their native range[32], promoting their introduction in areas where they

have not historically occurred ("assisted colonization"[35,75]) might be a viable conservation option, provided they do not significantly harm local communities. Similarly, when long-established introduced populations of LMH have limited impacts on native biodiversity but belong to species facing extinction, their presence might be tolerated or even enhanced as a part of a broader conservation strategy[76]. Species introduction, however, requires caution, as threatened species can not only establish alien populations successfully[77], but they can also pose a threat to local species, a phenomenon known as the conservation-invasion paradox[78,79]. The EICAT framework can be valuable for assessing such risks prior to introduction, as demonstrated in a recent expert elicitation effort that identified the site with the lowest risk across candidate locations for the conservation translocation of the extinct-in-the-wild sihek (Guam kingfisher; *Todiramphus cinnamominus*)[80]. By exploring which threatened mammal species have established alien populations worldwide, Tedeschi and colleagues[81] recently identified six species that were also included in our analysis. While one of them, the Indian hog deer (*Axis porcinus*), is characterized by only (weak) positive impacts, the other five (*Elephas maximus*, *Ammotragus lervia*, *Rangifer tarandus*, *Rusa timorensis*, *Rusa unicolor*) cause predominantly negative impacts once established, many of which have led to the decline or local extirpation of native species (Fig. 1).

The introduction of LMH outside their native ranges has additionally been suggested as a suitable strategy for restoring top-down trophic interactions and associated trophic cascades lost during the pleistocenic and holocenic human-mediated extinctions[82,83]. Such a strategy, commonly referred to as trophic rewilding[34], falls under the broader concept of rewilding, defined as "the process of rebuilding, following major human disturbance, a natural ecosystem by restoring natural processes and the complete or near-complete food web at all trophic levels, resulting in a self-sustaining and resilient ecosystem with biota that would have been present had the disturbance not occurred[84]. In support of trophic rewilding, a recent meta-analysis found that introduced LMH impact vegetation abundance or diversity similarly to native LMH, even on islands[85]. Future studies could employ the EICAT(+) frameworks to compare bidirectional impacts of introduced and native LMH populations, similarly to what has previously been done to investigate the negative impacts of bamboos[86], bark beetles[87] and marine fishes[88] from different biogeographic origins. While such comparisons can provide valuable insights, they do not alter our findings that by introducing LMH beyond their native ranges, humans have triggered predominantly, although not exclusively, negative impacts on native biodiversity. These impacts are particularly pronounced on islands and affect not only native plants but also higher trophic levels, suggesting that average impacts and community metrics, often measured only at one trophic level, may overlook the detailed patterns of biodiversity alteration. Decisions about the introduction or removal of alien LMH for conservation purposes, including assisted colonization, rewilding and eradication should therefore involve a careful risk assessment that considers the local context[84,89] and identifies winners and losers of anthropogenic interventions. Similarly, EICAT(+) data should not directly lead to management measures, but rather be used to inform local and national decision-making procedures on introduced LHM and other taxa[80], alongside socio-economic considerations[90], ethical trade-offs[91] and clarity in conservation goals.

## Methods

### LMH as a study system
LMH have important effects on terrestrial ecosystems by causing disturbances, consuming low-nutrient vegetation and dispersing plant propagules and nutrients[92]. They have an intermediate position within the food chain and were frequently introduced to both insular and continental sites[6,67]. Among the currently recognized 286 modern species of LMH (Cetartiodactyla, Perissodactyla, Proboscidea), including wild and domesticated forms[93], 66 species from six families have established alien populations according to the IUCN Global Register of Introduced and Invasive Species (http://www.griis.org). The impacts of their introductions outside their native ranges are controversial[85,94], but a systematic comparison of their positive and negative impacts on native species is currently lacking.

### Impact assessment frameworks
As measurable changes (decreases or increases) to ecosystem attributes[8], ecological impacts can be classified based on their direction, i.e., be distinguished between negative and positive impacts[21]. The interpretation of such impacts, however, might entail subjectivity because it depends on the selection of the ecosystem attributes that are measured—whether they are e.g., species, populations, individuals, genes, or abiotic ecosystem attributes—a choice guided by study purpose, feasibility and convention, but also by values and interests[31]. Values and interests also play a role in determining whether impacts are perceived as detrimental or beneficial to nature or people[20,21], thereby making it challenging to reach a consensus regarding the interpretation of impact direction[22,29]. Here, we consider native biodiversity as the entity of conservation concern[36] and use the assessments of its status as the baseline for evaluating alien species' impact. This approach allows us to classify the impacts of alien species discerning whether they pose detriments or offer benefits to local populations of native species, analogously to what is outlined in the recent global Thematic Assessment Report on Invasive Alien Species and their Control of the Intergovernmental Science-Policy Platform on Biodiversity and Ecosystem Services (IPBES)[95]. Detrimental and beneficial impacts are measured as decreasing and increasing changes to specific attributes of local populations of native species, such as performance of individuals, population size and area of occupancy, thus aligning with arithmetically defined negative and positive impacts[21,31]. To classify the magnitude of these negative and positive impacts, we have used the IUCN EICAT and EICAT+ frameworks[30,31,96]. Both frameworks use a semi-quantitative five-tier classification of impact magnitudes (ranging from "Minimal" to "Massive") based on a common set of attributes (Table 5). While impacts classified as "Minimal" and "Minor" distinguish whether changes in the performance of native individuals are detected, the other categories relate to changes at the population level. Specifically, "Moderate" impacts are assigned for changes in population size, while "Major" and "Massive" impacts relate to changes in the area of occupancy through extinction or re-establishment/extinction prevention of a local population (Table 5). Following the approach adopted by the IUCN[30], which classifies negative impacts on populations of native species (Moderate, Major, Massive) as "harmful", in contrast to impacts that do not involve changes in population size (Minimal, Minor), we further categorized the magnitude of both positive and negative impacts as either "strong" or "weak" (Table 5).

When the impact of a species could not be classified due to insufficient data, the species was classified as Data Deficient (DD) and not used for statistical analysis. Note that cases assigned as Data Deficient (DD) differ from Minimal impacts (MC/ML + ), where the study design would have allowed discovering impacts, but none were found. For each impact classified by both EICAT and EICAT+ we assigned a specific impact mechanism, with ten mechanisms for EICAT and eight for EICAT + [30,31] (Supplementary Table 1). Mechanisms were classified as either direct (when the alien species directly affected a native species, e.g., through competing for common resources or serving as food) or indirect (when the impact on the native species is indirect through changing another species or ecosystem property, e.g., transmitting a disease or suppressing a dominant competitor). To express the uncertainty associated with the accuracy of the assigned impact magnitudes, the assessor included a confidence level (Low, Medium, High) to evaluate how likely the assigned impact magnitude

**Table 5 | Criteria used to assess impact magnitudes in EICAT and EICAT+ and to categorize impact magnitude as "weak" or "strong"[30,31]**

| IUCN EICAT<br>Levels of impact magnitude for negative impacts | EICAT +<br>Levels of impact magnitude for positive impacts | Detailed criteria for defining negative/positive impacts as measurable decreases/increases in native species attributes | Impact magnitude (level of organization) |
|---|---|---|---|
| Minimal Concern (MC) | Minimal positive impact (ML +) | The alien taxon causes negligible decreases/increases in the performance of native individuals (i.e. their capacity to survive, gather resources, grow, or reproduce) | Weak (individual level) |
| Minor impact (MN) | Minor positive impact (MN +) | The alien taxon causes decreases/increases in the performance of native individuals, but no decrease/increase in the native population size. | |
| Moderate impact (MO) | Moderate positive impact (MO +) | The alien taxon causes a decrease/increase in the native population size, but no decrease/increase in the area of occupancy (through extinction/re-establishment or extinction prevention of a local population). | Strong (population level) |
| Major impact (MR) | Major positive impact (MR +) | The alien taxon causes a "reversible" decrease/increase in the area of occupancy (through extinction/re-establishment or extinction prevention of a local population). Reversible impacts are those which disappear after the removal of the alien taxon. | |
| Massive impact (MV) | Massive positive impact (MV +) | The alien taxon causes an "irreversible" decrease/increase in the area of occupancy (through extinction/re-establishment or extinction prevention of a local population). Irreversible impacts are those which do not disappear after the removal of the alien taxon. | |

reflects the true impact. This evaluation considers data type and quality, study design, spatial and temporal scale, presence of confounding effects, and the overall coherence of evidence as key factor in determining whether the true impacts may be higher, lower, or both compared to the assigned impact. For further information on assigning confidence levels see IUCN[30,96], and Probert et al[97].

## Collection of impact reports

We followed the search protocol described by Evans et al[98]. to collect the data and built upon the work of Volery et al[6]. by incorporating positive impacts of alien LMH on native species. The data sources were obtained by conducting a search using the following terms ('invasive' OR 'invasive species' OR 'introduced species' OR 'introduced' OR 'alien' OR 'non-native' OR 'non-indigenous' OR 'feral' OR 'exotic' OR 'positive impact' OR ' beneficial' OR 'benefit' OR 'positive effect' AND '[scientific name of the alien species]') in the online database Google Scholar (https://scholar.google.com) including articles published in scientific journals as well as gray literature, such as conference abstracts, governmental papers, and private sector research. Similar to Volery et al[6]. a literature review was performed for all 66 alien LMH species that have established populations outside their native range. In full accordance with the terminology adopted by the IUCN[30], we used the term "alien" for species, subspecies or breed, moved intentionally or unintentionally by human activities beyond the limits of their native geographic range, or that have resulted from breeding or hybridization and have been released into areas where they do not naturally occur. Conversely, the term "native" refers to taxa that have originated in a given area without human involvement or that have arrived without human involvement from an area in which they were native[99]. Data sources containing observed impacts of an alien LMH on a native population were selected based on the evaluation of the title, abstract, and content of the first 100 records found. Additionally, we followed up all references to other data sources with observed impacts in the selected papers until no additional impact records were found. The references gathered for negative impacts by Volery and coworkers[6] were cross-checked to identify any additional positive impacts. Only observed impacts were included for classification, while potential or inferred impacts were not considered, in line with the guidelines of the frameworks used[30,31,96].

Each impact observation recorded refers to a specific alien LMH species at a specific location and year, along with one impacted native species, the assigned impact magnitude, mechanism (direct or indirect; see Supplementary Table 1), and associated confidence level. Additional information, such as the reference of the impact observation, year of impact (publication year when no specific year was given in the report), taxonomy and trophic level of the impacted native species (decomposer, producer, primary consumer, and secondary consumer/omnivore), geographical details (including precise coordinates, country's sub-unit such as district, state, region or county, country, continent, mainland or island), assessor ID (i.e., the person sourcing and assessing impact observations under EICAT/+), assessment date, and reviewer ID (i.e., the person reviewing the data assessed under EICAT/+), were included as supplementary details. In line with IUCN recommendations[30,96], all negative impacts were cross-checked for consistency in impact magnitude and confidence by at least one reviewer[6]. All positive impacts were scored by Z.B.M. and cross-checked by G.V.

## Statistical analysis

All analyses were conducted in R (version 4.3.2). One-sided paired sign tests were conducted to investigate whether negative impacts were consistently more numerous than positive impacts in species and reports exhibiting bidirectional impact observations, using the package "BSDMA" (version 1.2.2)[100]. Pairwise z-tests were conducted to investigate whether the proportion of impact observations assigned with low, medium, and high confidence differs between negative and positive impacts at each level of impact magnitude. Generalized linear mixed-effect models (GLMMs) with binomial error distribution were built using the package "lme4" (version 1.1.35.1)[101] to test the effects of multiple predictors on impact magnitude, or more precisely, the probability of an alien LMH species causing a strong impact on native biodiversity. The response variable was the impact magnitude coded as "0" for "weak" and "1" for "strong" impacts, for both positive and negative impacts. The predictors coded as fixed effects were impact direction (positive vs. negative), the trophic level of the impacted species (four categories), the impact location (island vs. mainland), reporting impact year (scaled to 0 mean and 1 SD[102]), mechanism types (direct vs. indirect), and all 2-way interactions of the previous variables with impact direction. The Report ID and alien species name were included as random effects to account for pseudoreplication resulting from multiple observations from the same report and/or the same alien species.

Models with all different combinations of the fixed effects were fitted by maximum likelihood with the Laplace approximation. Models

were ranked based on their corrected Akaike Information Criterion (AICc). For the best-fitting models ($\Delta$AICc <6[103], we estimated the relative importance (sum of Akaike weights $\omega$[104]) of each variable and interaction (fixed effects) in the set of models to identify predictors with adequate explanatory power (>0.5). The evidence ratio of each model i ($\omega_1/\omega_i$, where $\omega_1$ is the Akaike weight of the best-fitting model) was also calculated to estimate the likelihood of each model being the most appropriate representation of the underlying data. While higher ratios indicate stronger evidence in favor of the best-fitting model, lower ratios suggest comparatively stronger support for alternative models. The most supported model, used for pairwise comparisons and graphical representations, was chosen as the model retaining all predictors with adequate explanatory power (relative importance > 0.5) among the models considered equally plausible (evidence ratio <2). To verify that the assumptions of the selected model were not violated, we used the R package "DHARMa" (version 0.4.6)[105] to examine the normality of the residuals with a QQ-plot and test for the presence of overdispersion and outliers ($p = 0.456$, $p = 1$).

GLMMs with binomial error distribution were additionally constructed to test whether weak and strong impacts were assigned with different levels of confidence, and to examine how the assignment of confidence varies by direction and over the years. By using magnitude as a response variable, the predictors coded as fixed effects were impact direction (positive vs. negative), confidence (three categories: low, medium and high), reporting impact year (scaled to 0 mean and 1SD), and all 2-way interactions of the previous variables. The Report ID and alien species name were included as random effects to account for pseudoreplication resulting from multiple observations from the same report and/or the same alien species. We hypothesized that strong impacts were assigned with higher confidence than weak impacts, while we did not expect any difference in confidence when assigning impact magnitude between positive and negative impacts. We also hypothesized that strong impacts assigned with high confidence were reported first. Consequently, we expected that impact magnitude would decline more quickly in impacts classified with high confidence in comparison with those assigned with lower degrees of confidence.

Tukey's pairwise multiple comparisons were performed to identify significant differences among levels of categorical variables using the *emmeans(), emmip()* and *emtrends()* functions from the package "emmeans" (version 1.8.6)[106]. Figures were plotted using the packages "ggplot2" (version 3.4.2)[107] and "sjPlot" (version 2.8.15)[108].

### Reporting summary

Further information on research design is available in the Nature Portfolio Reporting Summary linked to this article.

## Data availability

The complete dataset assembled and used in this study is provided in the Supplementary Data 1. All intermediate datasets used to conduct the analyses and to generate the figures and results are publicly available via Figshare at https://doi.org/10.6084/m9.figshare.28046465.v1.

## Code availability

The R code used to conduct the analyses and to generate the figures and results is publicly available via Figshare at https://doi.org/10.6084/m9.figshare.28046465.v1.

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

## Acknowledgements

This research was supported by the 2017–2018 Belmont Forum and BiodivERsA joint call under the BiodivScen ERA-Net COFUND program through the InvasiBES project, and by the Swiss National Science Foundation (SNSF) through grants 31003A_179491, 31BD30_184114 and IC00I0_231475 awarded to S.B.

## Author contributions

Z.B.-M., S.B., and G.V. conceived the study. Z.B.-M. collected the data and led the data analysis, with supervision by S.B. and G.V. G.V. led the manuscript writing, with significant contributions from Z.B.-M. and S.B., and prepared the figures. All authors have read and approved the final version of the manuscript for submission.

## Competing interests

The authors declare that they have no competing interests.
