## [Peer Review file · Nature Communications]

Harms of introduced large herbivores outweigh benefits to native biodiversity

Corresponding Author: Dr Giovanni Vimercati

Version 0:

Reviewer comments:

Reviewer #1

(Remarks to the Author)

Dear authors and editors,

I appreciate the opportunity to review Bescond-Michel et al.'s manuscript on the effects of introduced large herbivores. The authors employ the IUCN's EICAT and EICAT+ methodology to evaluate the positive and negative effects of large herbivores on native species. The manuscript is very well written and the authors were clearly very thorough and meticulous in their work.

Unfortunately, and with utmost respect to the authors' efforts here, I think there are fundamental conceptual flaws that make EICAT and EICAT+ useless for understanding the effects of introduced species. I am sorry to play the role of Reviewer #2.

EICAT and EICAT+ (henceforth just EICAT) are a method of vote counting to evaluate the effects of introduced species. As such, EICAT involves tallying impacts that reduce fitness or abundance of native species, which are considered 'negative' effects and impacts that increase fitness or abundance of native species, which are considered 'positive' effects.

The problem with EICAT is that it a priori assumes that introduced and native species are biologically distinct, as if you could tell whether a species is native or introduced based on their measurable empirical effects on ecosystems. Most evidence suggests that you cannot (in general: Boltovskoy et al. 2021, with large herbivores Lundgren et al. 2024). The EICAT criteria, if applied to cherished native species, would find, for example, that otters have negative effects by reducing sea urchin populations; that elephants have negative effects by damaging trees; that wolves have negative effects by reducing elk populations. And so on.

The literature is replete with many examples of native large herbivores causing very similar effects as introduced ones, including as the most serious impacts documented in the present study. Giraffe reintroduction in South Africa led to the local extirpation of several tree species (Bond and Loffell 2001); native deer can heavily suppress forest herb diversity and abundance (VanderMolen and Webster 2021); native large herbivores in general reduce small mammal abundance and diversity through competition (Daskin and Pringle 2016, Afonso et al. 2023); and endangered Key deer are implicated in the severe decline and endangerment of plant communities in the Florida Keys (Univ. of South Florida, Tampa, USA et al. 2006). Indeed, in some ways, these cases seem more rigorous and provocative than the 4 most severe and highest quality 'negative' impact studies the authors cite in the present manuscript (moderate impacts with medium to high confidence) (see Supplementary data).

The EICAT framework seems to assume that native ecosystems exist in some kind of Edenic harmony where native species do not reduce the abundance of their resources or their competitors. But it should be obvious that there is no way for any organism to not have a negative effect. It is literally impossible for any species to not have negative impacts as long as they take up space and have a metabolism (i.e., are alive) .

The EICAT framework, even with the EICAT+ amendment (which is designed to be make the process more neutral and open to 'positive' effects), is thus stacked to almost always find that introduced species have predominantly negative effects because the direct effects of organisms on their resource bases are usually negative and are much easier and cheaper to measure than indirect downstream facilitative effects. The authors' results acknowledge this in the Abstract: "Most positive

impacts were caused indirectly through changes in species interactions and ecosystem properties, often following negative impacts on native plants through herbivory and disturbance.”

In order to claim that the predominantly negative effects of introduced herbivores are a function of nativeness per se, the authors need to have a proper control: native herbivores. After all, rhinos dramatically reduce vegetation cover (easily measured negative effect) but if you were to study the system long enough you would also see that doing so facilitates smaller grazers (positive effect) and reduces wildfire (positive effect) (Waldram et al. 2008). Likewise, otter consumption of urchins (negative) protects kelp forests (positive) (Estes and Palmisano 1974), wolf predation of elk (negative) protects riparian vegetation (positive) (Ripple et al. 2001), the exclusion of large herbivores in East Africa leads to a doubling in small mammals (a negative effect of herbivores) and to dramatic increases in tick densities and thus potentially to increases in tick-borne disease (which is positive for the ticks but negative for the disease recipients, (Titcomb et al. 2017)); and elephant induced damage and mortality of trees (negative) promotes increases understory biomass, reduces herbivory, and promotes open habitats (which is positive for some species and negative for others (Coverdale et al. 2016, Gordon et al. 2023). These are stories of wonderfully complex and nuanced ecology, but introduced species are held to a different and highly moralized standard.

As such, the EICAT framework fails to have proper controls to make any claim that the effects described have anything to do with the nativeness status of the organisms in question (as implied by the way these results are communicated, including in the title of the manuscript under consideration here).

The balance of nature mythology embedded within EICAT seems to persist only in the realm of invasion biology and creates a rigged and moralistic way to evaluate the effects of introduced species, one that cannot be supported scientifically. If we were to apply the EICAT criteria to all species, native or introduced, we would find that all species are horrible and must be destroyed. This is what philosophers call a tautology: a self-fulfilling truism that cannot be falsified and has thus left the realm of science.

It's definitely true that some introduced herbivores can cause extinctions or massive range declines, which is especially likely on islands. The authors of the present manuscript did indeed find limited evidence of the local extirpation of a few species in 4 studies (of which several returned following the construction of exclosures). These of course are topics of conservation concern, as when native large herbivores cause similar impacts (e.g., the effects of endangered Key deer or reintroduced giraffe). However, these 4 studies hardly support the central title and claim of this manuscript.

Instead, most of the negative evidence compiled in this article is exactly what you'd expect herbivores to do by definition of being herbivores: reducing the abundance and cover/biomass of their food resources (though this is actually difficult to parse from their data). Reducing abundance is not the same thing as causing extinctions. However, it should also be noted that local extirpations of herbivory sensitive species should be expected: many plants are restricted to spatial or temporal refugia due to native herbivores (e.g., succulent euphorbs in Africa (Cowling et al. 2010), the trees in Bond's giraffe study, etc).

The authors' chief conclusion is to caution against reintroducing extinct large herbivores or proxies. However, the argument for rewilding large herbivores is to restore their top-down influences on ecosystems in regulating vegetation, especially dominant plant species (Svenning et al. 2019). This top-down trophic regulation was a part of Earth's history for ~55 million years until human-caused extinctions ~45,000-600 years ago (Svenning et al. 2024) and appears to play a strong role in biodiversity maintenance at broad scales, at reducing wildfire frequency and severity, at maintaining nutrient and seed dispersal, creating open habitats, and so on (Doughty et al. 2016, Malhi et al. 2016, Karp et al. 2021). However, the way EICAT is designed, these same direct top-down influences are called 'negative' effects. As such, I do not see how the results of this paper have any bearing on recommendations for rewilding efforts. The real issue is a disagreement in values about what is 'good' or not in the world: something that science itself cannot assess ((Slobodkin 2001, Sagoff 2020).

Finally, 50% of introduced large herbivores are threatened or extinct in their native ranges (which the authors do cite). Eradications could cause the extinction of species in the wild. These animals are also known to be intelligent and cultural beings with strong emotional ties to each other and they had no choice but to be entangled in human conquest and empire building. Instead of calling them 'invaders' we could be equally problematic and anthropomorphic but more accurately so and call these animals 'escaped slaves'. Would we ask the same questions or treat their impacts similarly?

I wish I had more constructive feedback to provide but I just do not see how the simplistic moral mathematics of EICAT, which lacks a proper control to make the claims it makes, can help us understand and respond appropriately to radical environmental change. I understand that EICAT and this type of reasoning is pervasive but that does not make it scientific. As far as I can tell, the authors' results simply suggest that introduced large herbivores are, indeed, herbivores.

As such, I cannot support the publication of this article in as fine a journal as Nature Communications. If the authors had done something far more difficult, such as doing an EICAT analysis of the effects of native herbivores as well (the proper control to make any claim about the effects of introduced herbivores) then this article would be fascinating and a valuable contribution to the literature. I know that is a lot to ask though.

Afonso, B. C., L. M. Rosalino, J. Henriques, R. Tinoco Torres, J. Wauters, and J. Carvalho. 2023. The effects of wild ungulates on small mammals: a systematic review and meta-analysis. *Mammal review*.

Boltovskoy, D., N. M. Correa, L. E. Burlakova, A. Y. Karatayev, E. V. Thuesen, F. Sylvester, and E. M. Paolucci. 2021. Traits

and impacts of introduced species: a quantitative review of meta-analyses. *Hydrobiologia* 848:2225–2258.

Bond, W. J., and D. Loffell. 2001. Introduction of giraffe changes acacia distribution in a South African savanna. *African journal of ecology* 39:286–294.

Coverdale, T. C., T. R. Kartzinel, K. L. Grabowski, R. K. Shriver, A. A. Hassan, J. R. Goheen, T. M. Palmer, and R. M. Pringle. 2016. Elephants in the understory: opposing direct and indirect effects of consumption and ecosystem engineering by megaherbivores. *Ecology* 97:3219–3230.

Cowling, R. M., A. Kamineth, M. Difford, and E. E. Campbell. 2010. Contemporary and historical impacts of megaherbivores on the population structure of tree euphorbias in South African subtropical thicket. *African journal of ecology* 48:135–145.

Daskin, J. H., and R. M. Pringle. 2016. Does primary productivity modulate the indirect effects of large herbivores? A global meta-analysis. *The Journal of animal ecology* 85:857–868.

Doughty, C. E., J. Roman, S. Faurby, A. Wolf, A. Haque, E. S. Bakker, Y. Malhi, J. B. Dunning Jr, and J.-C. Svenning. 2016. Global nutrient transport in a world of giants. *Proceedings of the National Academy of Sciences of the United States of America* 113:868–873.

Estes, J. A., and J. F. Palmisano. 1974. Sea otters: their role in structuring nearshore communities. *Science (New York, N.Y.)* 185:1058–1060.

Gordon, C. E., M. Greve, M. Henley, A. Bedetti, P. Allin, and J.-C. Svenning. 2023. Elephant rewilding affects landscape openness and fauna habitat across a 92-year period. *Ecological applications: a publication of the Ecological Society of America* 33:e2810.

Karp, A. T., J. T. Faith, J. R. Marlon, and A. C. Staver. 2021. Global response of fire activity to late Quaternary grazer extinctions. *Science* 374:1145–1148.

Lundgren, E. J., J. Bergman, J. Trepel, and E. le Roux. 2024. Functional traits—not nativeness—shape the effects of large mammalian herbivores on plant communities. *Science*.

Malhi, Y., C. E. Doughty, M. Galetti, F. A. Smith, J.-C. Svenning, and J. W. Terborgh. 2016. Megafauna and ecosystem function from the Pleistocene to the Anthropocene. *Proceedings of the National Academy of Sciences of the United States of America* 113:838–846.

Ripple, W. J., E. J. Larsen, R. A. Renkin, and D. W. Smith. 2001. Trophic cascades among wolves, elk and aspen on Yellowstone National Park's northern range. *Biological conservation* 102:227–234.

Sagoff, M. 2020. Fact and value in invasion biology. *Conservation biology: the journal of the Society for Conservation Biology* 34:581–588.

Slobodkin, L. 2001. The good, the bad and the reified. *Evol. Ecol. Res.* 3:91–105.

Svenning, J. C., R. T. Lemoine, and J. Bergman. 2024. The late-Quaternary megafauna extinctions: patterns, causes, ecological consequences, and implications for ecosystem management in the Anthropocene. *Cambridge Prisms* 2.

Svenning, J.-C., M. Munk, and A. Schweiger. 2019. Trophic rewilding: ecological restoration of top-down interactions to promote self-regulating biodiverse ecosystems. in N. Pettorelli, S. M. Durant, and J. T. du Toit, editors. *Rewilding*. Cambridge University Press.

Titcomb, G., B. F. Allan, T. Ainsworth, L. Henson, T. Hedlund, R. M. Pringle, T. M. Palmer, L. Njoroge, M. G. Campana, R. C. Fleischer, J. N. Mantas, and H. S. Young. 2017. Interacting effects of wildlife loss and climate on ticks and tick-borne disease. *Proceedings. Biological sciences* 284.

Barrett, P. Stiling, and Univ. of South Florida, Tampa, USA. 2006. Impacts of endangered Key deer herbivory on imperiled pine rocklands: a conservation dilemma? *Animal biodiversity and conservation* 29:165–178.

VanderMolen, M. S., and C. R. Webster. 2021. Influence of deer herbivory on regeneration dynamics and gap capture in experimental gaps, 18 years post-harvest. *Forest ecology and management* 501:119675.

Waldram, M. S., W. J. Bond, and W. D. Stock. 2008. Ecological engineering by a mega-grazer: White rhino impacts on a south African Savanna. *Ecosystems* 11:101–112.

Reviewer #2

(Remarks to the Author)

This is an interesting analysis of the impacts of invasive alien large herbivores. I think the paper could be a useful

contribution to our understanding of the subject, but there are aspects of the analysis and discussion that need to be addressed.

Main comments

(1) Confirmation bias/controlling for effort. A significant concern I have about the analysis is the risk of confirmation bias in the underlying dataset. In the context of the global biodiversity, many scientists are undertaking research to identify negative impacts. This is perfectly valid because it is negative impacts that do harm to ecosystems. However, this means there may be more scientists looking/designing monitoring programs/experiments to identify negative impacts than positive impacts. In other words there is risk of confirmation bias when these data are used for comparative studies such as this. This means the dataset may contain differential effort. Indeed between lines 432 – 436, the authors point out the increase in interest in positive impacts of invasive species. This implicitly supports my concern that there has been less effort looking for positive impacts. The comparison of positive versus negative impacts is central to this manuscript. Please can the authors outline whether or not they controlled for this potential bias/differential effort? If not, I think they need to look at doing so. Further, I think this issue of potential bias in the underlying dataset should be explored in the Discussion section.

(2) Defining nativeness. The authors don't really define what they mean by native/nativeness. I think this needs to be addressed and discussed. One example they use is of subspecies of *Cervus elaphus*. This is still the same species, and I am unconvinced that this is splitting the definition of nativeness too far for a broad-scale analysis like this. By definition as the same species, it is arguable whether different sub-species can be functionally considered "alien". Further, given the propensity for taxonomists to split and then lump sub-species, such definitions are fluid. Also, due to the fragmentation of populations, many geneticists consider it good practice to mix sub-populations to enhance genetic diversity and adaptive capacity; especially where inbreeding has occurred.

Another consideration is the "refugee species concept" (see the work of Graham Kerley and colleagues). This is where species ranges have been drastically reduced over time, and extant populations have been confined to refuge habitat that may not be optimal. However, such sub-optimal habitat often become the "stereotype" for that species i.e. where it is considered "native". There are multiple issues with this for conservation, but a key one is that we may underestimate the former range of a species i.e. where we would consider it native. Under the criteria used in this manuscript, a species could be considered "alien" in places where it is not to have formerly occurred. Similarly, under climate change scenarios, the potential niche of a species may change, and it is legitimate to consider where a species might thrive outside our current definition of its range. The authors recognise this, but it does raise the question about whether seeing "alienness" as a binary concept is too rigid.

Another example of the complexities, is where a species has become extinct but it had important ecological functions for the ecosystem that it occupied. It is reasonable to consider niche substitutes as part of a rewilding program. Those species that are evolutionarily and geographically closer are probably less alien than those that are further away. I think alienness is therefore more of a continuum i.e. some species are more alien than others. I agree with the authors that caution should be exercised with such rewilding, but there is a risk that taking a binary view of impacts may miss opportunities for positive conservation outcomes. It would be good to see more of a discussion of these nuances around definitions of what an alien species is, and what the risks are.

(3) Novel ecosystems and coexistence. While I recognise that the aim of this paper is to try and quantify the problem/impacts of native species, how does this apply to novel ecosystems? Many ecosystems are modified to a greater or lesser extent by humans (this occurs on a continuum). Such modified ecosystems contain novel mixes of species. The focus of impact on native species while understandable, may miss the reality in many parts of the world. Many invasive species are very hard to control, and total eradication may be impossible. Therefore, managers in many situations have no choice but to manage for some form of uneasy coexistence between native and non-native ecosystem elements. I think it would be good to see more acknowledgement of this complexity and nuance when considering the implications of the findings of this analysis.

Detailed comments.

Line 18 – comma after common and magnitude

- See Main Comments regarding comparison of negative and positive effects

Line 72 – "population-level changes" – to native species?

Line 80 and 82 – see Main Comments. What is meant by "native" needs to be defined.

Line 115 – "conservation perspective". Please define what is meant by this?

Line 154 – 155 - Please expand more i.e. given the importance of this to the analysis, I don't think it is enough to refer readers to another document.

Line 166 - I suggest adding something like "that have established populations (see above" after species. The analysis is focussed on the 66 species that have established out of a total of 286. Please check whole manuscript and ensure clarity about where referring to established species.

Line 169-171 – See Main Comments. There is risk of bias/confirmation bias towards negative impacts in the dataset that needs to be addressed.

Line 189 – See Main Comments regarding risk of bias in dataset.

Line 207 – Please provide examples of conspecifics.

Line 233 and 236 – "were" rather than "are" – please check tenses throughout manuscript.

Line 236 – "expected".

Line 246 - See Main Comments regarding risk of bias in dataset.

Line 250 - See Main Comments regarding risk of bias in dataset.

Line 261 - See Main Comments regarding risk of bias in dataset.

Line 265 – 268 - See Main Comments regarding risk of bias in dataset.

Line 288 – 289 - See Main Comments regarding risk of bias in dataset.

Line 346 – 347 - See Main Comments regarding risk of bias in dataset.

Line 411 - See Main Comments regarding risk of bias in dataset. This needs to be discussed in the Discussion as a potential limitation.

Line 432 – 436 - See Main Comments regarding risk of bias in dataset. This reference to “recent” popularity implicitly implies that this may have been underreported/studied before this popularity.

Line 482 – Wild boar are omnivores, so are quite distinct from the majority of large herbivores that are obligate herbivores. Given that, is it valid to refer to “most predation events” on islands in a general sense, when it refers to only an omnivore? Should wild boar be included in this analysis? To what extent is this omnivore influencing results in the rest of the analysis and dependent conclusions?

Line 493 – as above, omnivorous wild boar are lumped with obligate herbivores.

Line 514 – 516 – What is the risk that invasive alien herbivores could help inflate populations of both native and invasive predators that could then enhance impact on native prey species through prey swapping? I think this needs to be discussed.

Line 566 – Please define rewilding and provide a reference. The following reference has the most recent definition:

Carver, S., Convery, I., Hawkins, S., Beyers, R., Eagle, A., Kun, Z., Van Maanen, E., Cao, Y., Fisher, M., Edwards, S.R. and Nelson, C., 2021. Guiding principles for rewilding. *Conservation Biology*, 35(6), pp.1882-1893.

However, if you are referring to “trophic rewilding” see:

Svenning, J.C., Pedersen, P.B., Donlan, C.J., Ejrnæs, R., Faurby, S., Galetti, M., Hansen, D.M., Sandel, B., Sandom, C.J., Terborgh, J.W. and Vera, F.W., 2016. Science for a wilder Anthropocene: Synthesis and future directions for trophic rewilding research. *Proceedings of the National Academy of Sciences*, 113(4), pp.898-906.

Reviewer #3

(Remarks to the Author)

Comments on Bescond-Miochel et al.

In a novel quantitative synthesis, the authors have found that introductions of large mammalian herbivores outside their native range have “both harmed and benefited local native biodiversity, but negative consequences have largely surpassed positive outcomes, both in frequency and magnitude” and that such “negative impacts are more numerous, larger and often precede positive impact.” These results add new evidence to a debate that has unfolded on the opinion pages of journals and in the popular media, in which some researchers have argued that alien species impacts are biased toward negative effects. However, as the authors have correctly asserted (lines 52-53), there has been no rigorous study demonstrating that the negative impacts of alien species on native biodiversity are biased. For example, there is abundant documented evidence of the major role of invasions as a driver of global extinctions, especially on islands worldwide, but no one has shown that these severe impacts are balanced out by positive effects.

Another noteworthy finding is that introduced mammalian herbivores that have had positive population-level impacts have done so mainly through indirect interactions with other species or disruptions to ecosystem properties that favor some certain taxa. I expect that this finding will add substance to the debate concerning the costs and benefits of alien megafaunal introductions – such as those at the center of proposed assisted colonization and rewilding schemes.

In summary, this study is a welcome and timely addition to the literature. I am confident that these findings will be widely cited, and probably debated, by ecologists and conservation biologists.

Specific comments:

Discussion: Are there cases where positive impacts clearly resulted in an enhanced ecosystem service or where negative impacts resulted in a disrupted or degraded ecosystem service? I understand that this may be outside the scope of the study.

Discussion: Can you discuss whether any of the species that caused strong negative impacts (at any location where they have been introduced) are imperiled in their native ranges? (e.g., Barbary sheep, mouflon, others?)

Discussion: You could mention that a potentially large set of indirect non-trophic effects have been ignored because such effects require very detailed study to recognize. However, we do know they exist; for example pigs create 'wallows' - muddy depressions that fill with water - and thus create habitat for mosquitoes (which could be classified as an indirect 'positive effect' on an insect), which in Hawaii likely exacerbated transmission of avian malaria (a clearly devastating effect on native avifauna).

Lines 64-67: The authors could mention that species threatened or endangered in their native ranges can nonetheless become invasive where they have been introduced, owing to context dependencies (the Conservation-Invasion Paradox).

Line 82: “Ecological dynamics” - I suggest using the term “ecoevolutionary dynamics” as it explicitly includes evolutionary

context as a factor, and thus recognizes that disruption can arise for evolutionary mismatches (e.g., inadequate anti-herbivore defenses in native plants, lack of adaptation to disturbance from trampling, etc) and new interactions (e.g. facilitation of invasive plants).

Line 90: Re: Introduced species can have positive effects if they “serve as an important novel food resource for native consumers” - This would only be a significant benefit to native consumers if food resources are otherwise limiting or of lower energetic value.

Line 93: “Impact magnitude would decline over time” – In this context, how did you treat local extinctions as a negative impact? The impact magnitude is obviously large but of limited duration and ends when the native population is wiped out, unless you classify it distinctly as a legacy effect.

Lines 107-109: Perhaps emphasize here or elsewhere that several of the introduced species considered in this study might have significant positive or negative economic values that contrast with their ecological impacts; but those values are not being assessed here, otherwise species like hog deer (and others that pose a threat to crops) would have been ranked differently.

Lines 153-155: The cited paper by Volery et al offers a vague methodology for assessing confidence: "When the assessor did not find evidence indicating that the true impact magnitude is likely to be the assigned one, a Low confidence level was assigned." This begs the question, why was an impact magnitude assigned if no evidence existed in the first place?

Lines 233-238: Can you offer an ecological reason for an observed decline in impact (e.g. native predators learning to capture the introduced herbivore)?

Lines 258-261: “Species for which only positive impacts have been reported” include hog deer, Nilgai, red deer. Yet, according to the Global Invasive Species Database, negative ecological impacts of introduced red deer have been reported in South America (<https://www.iucngisd.org/gisd/species.php?sc=119>). Nilgai have been reported to damage mangroves in Texas (<https://scholarworks.utrgv.edu/etd/339/>). Perhaps I missed something, but does this imply that impacts are conservatively estimated in this study, or that some reports did not meet search criteria?

Lines 462-463: “Similar ecosystem changes from woodlands to grasslands (including ferns) were promoted by widespread alien deer species such as *Cervus elaphus*”. Although successional species (ferns) benefit from the introduced deer’s activities, they do so at the expense of previously established species in the woodland community. Then why are no negative impacts assigned to *C. elaphus* in Figure 1?

Version 1:

Reviewer comments:

Reviewer #2

(Remarks to the Author)

The manuscript is much improved, and I thank the authors for the amount of work they have done to engage with and address the reviewers' comments. While I think the manuscript could be an important contribution, I continue to have a number of concerns that I will outline below. I note the line numbers did not correspond - so I found it difficult to verify every point made.

(1) Bias. The authors have undertaken considerable work to address my concern about a bias towards studies focused on negative impacts. However, the changes they have made are essentially arguments of logic. If effects are compared, I think the bias/different effort needs to be controlled/accounted for in the modelling. This will give the reader more confidence in the comparisons.

(2) Nativeness - *Cervus elaphus*. I remain unconvinced by splitting impact to the sub-species level, nor that sub-species are necessarily alien. Many populations/ecologically significant units are an artifact of fragmentation of former connected clines. The classification of sub-species is constantly being reviewed. I think this distinction is not necessary in the context of the subject of impact. Also, I think managers would be alarmed if they have to think about sub-species x impact given all the factors they already have to manage at once.

(3) Refugee species concept (RSC) and nativeness. I don't think the authors fully recognise the potential impact of the RSC on their binary categorisation of native versus alien. A key point about RSC is that most ecologists and conservation biologists etc don't currently consider its effect and impacts on species distribution and habitat stereotypes. Thus 'relying on the authors' judgement' reinforces this. I reiterate that I think the implications of RSC on categorising alien or native is important.

Reviewer #3

(Remarks to the Author)

I have read the manuscript and the authors' responses, and I found them to have addressed my concerns quite effectively. I

am impressed by how the authors' handled the reviewers' comments in general.

Version 2:

Reviewer comments:

Reviewer #2

(Remarks to the Author)

I thank the authors for the extensive amount work they have undertaken in response to my comments. On the matters I raised, my key suggestions were to encourage the authors to discuss the caveats of their analysis. They have now done this very well, and also done additional analyses. This is excellent.

I thank the authors for resolving my point about *Cervus elaphus* sub-species. At the same time, I do take their excellent point about domestic and wild forms of Atlantic salmon. They may wish to add some words on scenarios where sub-species could be damaging if they wish to make a point about it - but I don't mind i.e. not essential from my perspective.

I thank the authors for the extra analysis regarding refugee species concept. This demonstrates the robustness of rigour of their work. I do slightly disagree that evidence is limited for RSC - for example see:

Smith, K.J., Pierson, J.C., Evans, M.J., Gordon, I.J. and Manning, A.D., 2024. Continental-scale identification and prioritisation of potential refugee species; a case study for rodents in Australia. *Ecography*, 2024(9), p.e07035.

DETAILED RESPONSES TO THE REVIEWERS' COMMENTS

Please find our detailed responses in red below, with all changes highlighted using track changes in the revised manuscript. Note that the line numbers refer to the revised manuscript.

Reviewer #1 (Remarks to the Author):

Dear authors and editors,

I appreciate the opportunity to review Bescond-Michel et al.'s manuscript on the effects of introduced large herbivores. The authors employ the IUCN's EICAT and EICAT+ methodology to evaluate the positive and negative effects of large herbivores on native species. The manuscript is very well written and the authors were clearly very thorough and meticulous in their work.

Unfortunately, and with utmost respect to the authors' efforts here, I think there are fundamental conceptual flaws that make EICAT and EICAT+ useless for understanding the effects of introduced species. I am sorry to play the role of Reviewer #2.

We appreciate the feedback provided and recognition of the thoroughness of our work. In the following, we outline why the EICAT(+) framework is a valuable tool for understanding the impacts of introduced species. We have addressed the reviewer's concerns in the revised manuscript and have clarified the misunderstandings regarding the framework's concept and utility. See our detailed responses below.

EICAT and EICAT+ (henceforth just EICAT) are a method of vote counting to evaluate the effects of introduced species. As such, EICAT involves tallying impacts that reduce fitness or abundance of native species, which are considered 'negative' effects and impacts that increase fitness or abundance of native species, which are considered 'positive' effects.

The problem with EICAT is that it a priori assumes that introduced and native species are biologically distinct, as if you could tell whether a species is native or introduced based on their measurable empirical effects on ecosystems. Most evidence suggests that you cannot (in general: Boltovskoy et al. 2021, with large herbivores Lundgren et al. 2024). The EICAT criteria, if applied to cherished native species, would find, for example, that otters have negative effects by reducing sea urchin populations; that elephants have negative effects by damaging trees; that wolves have negative effects by reducing elk populations. And so on.

The literature is replete with many examples of native large herbivores causing very similar effects as introduced ones, including as the most serious impacts documented in the present study. Giraffe reintroduction in South Africa led to the local extirpation of several tree species (Bond and Loffell 2001); native deer can heavily suppress forest herb diversity and abundance (VanderMolen and Webster 2021); native large herbivores in general reduce small mammal abundance and diversity through competition (Daskin and Pringle 2016, Afonso et al. 2023); and endangered Key deer are implicated in the severe decline and endangerment of plant communities in the Florida Keys (Univ. of South Florida, Tampa, USA et al. 2006). Indeed, in some ways, these cases seem more rigorous and provocative than the 4 most severe and highest quality 'negative' impact studies the authors cite in the present manuscript (moderate impacts with medium to high confidence) (see Supplementary data).

The EICAT framework seems to assume that native ecosystems exist in some kind of Edenic harmony where native species do not reduce the abundance of their resources or their competitors. But it should be obvious that there is no way for any organism to not have a negative effect. It is literally impossible for any species to not have negative impacts as long as they take up space and have a metabolism (i.e., are alive) .

EICAT(+) core assumptions - Although EICAT and its derivative (EICAT+) do assume that introduced and native populations are bio-geographically distinct entities, they do not assume that one could tell whether a species is native or introduced based on their measurable empirical effects on ecosystems. EICAT and EICAT+ provide a standardised, transparent, and objective framework for assessing to what extent a species introduced by humans into its non-native range impacts native species in the recipient environment. The notion that reductions in performance/abundance are classified as "negative" refers to the native species, i.e. if an introduced species reduces the performance of a native species EICAT classifies this as negative for the native species, and vice versa for increases in performance/abundance as positive for the native species. In other words, EICAT(+) provides a coherent foundation for measuring the impacts of the introduction of species into non-native ranges on native biodiversity, which is of widespread concern for conservation decisions.

Impact of species are not restricted to their introduced range - We fully acknowledge that impacts (as measurable changes) on native species are not limited to introduced species. As illustrated in the examples involving otters, elephants and wolves, among others, native species inherently have negative and positive impacts on other native species. Antagonistic interactions reduce the available resources or suitable habitats for certain species, shaping their distribution and abundance and confining their fundamental niches to narrower realised niches. Similarly, mutualistic and commensalistic interactions can expand the availability of resources or enhance habitat suitability, allowing species to occupy broader ecological niches, increase their abundance, and extend their distributions. Most ecologists would agree that any native species has specific negative and positive effects on other species in the community.

Such a scientific understanding forms the conceptual foundation of EICAT and EICAT+, which both describe a set of mechanisms through which introduced species alter the attributes (such as performance or abundance) of other species. Even native species re-introduced by humans in their historical range (e.g. giraffes in South Africa) or whose abundance or distribution were altered by human activities (e.g. via overexploitation or climate change) can have various impacts on some other native species.

More than just comparing native and introduced species – The EICAT methodology could even be used to explore how a native species affects other native species and/or would allow a comparison with introduced species. In a recent paper (Forgione, Bacher, Vimercati 2022 *Diversity and Distributions* 28(9), 1832-1849), we have demonstrated this by comparing impact magnitudes of native, range-expanding and introduced populations of bark beetles with the EICAT approach and found significant differences. Similar EICAT-based comparative studies were conducted on bamboos (Canavan et al. 2019) and marine species (Henry and Sorte 2021). These studies, that have now been cited at line 502, addressed research questions that are, however, fundamentally different from the one asked in our paper. Also, the conservation implications vary significantly depending on whether the goal is to reintroduce a native species, replace it with an alien species performing a similar ecological role (ecological replacement), or assess the impacts of a novel introduction of an alien species on existing native communities. The papers by Boltovskoy et al. 2021 and Lundgren et al. 2024 address the former conservation issues, while our paper specifically addresses the latter issue. While it is of scientific interest comparing the impact magnitude and direction between introduced and native taxa, this is not the only valid and scientifically meaningful question in the context of conservation or management. Restricting the focus of conservation solely to comparing the impacts of alien populations with those of ecologically 'equivalent' native species overlooks the key conservation objective of understanding how new species introductions affect existing native communities and ecosystems (see lines 503-506). Such a consideration might also be extended to reintroductions, particularly when the extirpation happened a long time ago, because while these anthropogenic efforts aim to restore native species, they too might have unforeseen ecological consequences and shift ecosystems towards an undesirable state.

The EICAT framework, even with the EICAT+ amendment (which is designed to be make the process more neutral and open to 'positive' effects), is thus stacked to almost always find that introduced species have predominantly negative effects because the direct effects of organisms on their resource bases are usually negative and are much easier and cheaper to measure than indirect downstream facilitative effects. The authors' results acknowledge this in the Abstract: "Most positive impacts were caused indirectly through changes in species interactions and ecosystem properties, often following negative impacts on native plants through herbivory and disturbance."

In our study we observed that negative impacts were predominantly direct while positive impacts were predominantly indirect. However, direct positive impact mechanisms (e.g.,

providing a novel food source) and indirect negative impacts (e.g. through altering abiotic conditions) also exist and have been documented (e.g. as systematically detailed in Vimercati et al. 2022), showing there is no a priori reason to expect negative impacts to be inherently easier to detect than positive ones nor that negative impacts must be larger than positive ones. The fact that both EICAT and EICAT+ describe a large range of direct and indirect impact mechanisms indicates that there is no fundamental bias in the frameworks toward categorizing impact direction. While it is sometime suggested that detecting negative impacts is easier and cheaper, we found at least 22 papers in which the authors selectively picked cases to highlight only specific positive impacts on a certain native species (see now line 320). This indicates that once these impacts were identified in the literature, EICAT+ enabled us to assess them; the low number of positive impacts compared to negative impacts is not due to a structural bias in the EICAT(+) frameworks, but rather reflects an inherent asymmetry in how alien LMH affect native biodiversity. Additionally, if direct impacts were indeed easier to identify, this should apply to both negative and positive impacts. For instance, indirect negative impacts often require complex study designs and advanced techniques, whereas the positive direct effects of introducing a novel food source on native predators are relatively straightforward to observe. The combined use of EICAT and EICAT+ allowed us to capture both cases, without a specific a priori expectation. We now discuss in a novel section (lines 302-334) whether other biases might have influenced our results.

In order to claim that the predominantly negative effects of introduced herbivores are a function of nativeness per se, the authors need to have a proper control: native herbivores. After all, rhinos dramatically reduce vegetation cover (easily measured negative effect) but if you were to study the system long enough you would also see that doing so facilitates smaller grazers (positive effect) and reduces wildfire (positive effect) (Waldrum et al. 2008). Likewise, otter consumption of urchins (negative) protects kelp forests (positive) (Estes and Palmisano 1974), wolf predation of elk (negative) protects riparian vegetation (positive) (Ripple et al. 2001), the exclusion of large herbivores in East Africa leads to a doubling in small mammals (a negative effect of herbivores) and to dramatic increases in tick densities and thus potentially to increases in tick-borne disease (which is positive for the ticks but negative for the disease recipients, (Titcomb et al. 2017)); and elephant induced damage and mortality of trees (negative) promotes increases understorey biomass, reduces herbivory, and promotes open habitats (which is positive for some species and negative for others (Coverdale et al. 2016, Gordon et al. 2023). These are stories of wonderfully complex and nuanced ecology, but introduced species are held to a different and highly moralized standard.

As such, the EICAT framework fails to have proper controls to make any claim that the effects described have anything to do with the nativeness status of the organisms in question (as implied by the way these results are communicated, including in the title of the manuscript under consideration here).

We do not claim that observed impacts are attributable to the nativeness status and we also do not claim that alien species have larger impacts than native species (which indeed would require a comparison as suggested above). We observe that the human-driven introduction of LMHs outside their native range generally has a negative impact on native communities, with more native species suffering than benefiting (lines 281-28 and 503-509). We also show that the magnitude of negative impacts outweighs the positive ones. This is measured in a transparent, standardised and objective way without involving any moral standards. We do not exclude that some large herbivores may have on balance more positive than negative impacts in certain ecosystems or might increase certain community metrics (e.g. species richness, diversity; see now lines 507-509). Such an evaluation would however require a different context-specific analysis and alternative methodological approaches, which were beyond the scope of our current study.

The balance of nature mythology embedded within EICAT seems to persist only in the realm of invasion biology and creates a rigged and moralistic way to evaluate the effects of introduced species, one that cannot be supported scientifically. If we were to apply the EICAT criteria to all species, native or introduced, we would find that all species are horrible and must be destroyed. This is what philosophers call a tautology: a self-fulfilling truism that cannot be falsified and has thus left the realm of science.

We believe the argumentation lacks scientific support and seems unfair. We do not abide by a “balance of nature mythology” and EICAT assessments do not lead to conclude that “all species, native or introduced, [...] are horrible and must be destroyed”. By considering and assessing positive impacts, we have aimed to identify which introduced species benefit native species and under which circumstances. Moreover, as clearly described in the IUCN Standard and accompanying papers, EICAT assessments do not directly lead to management recommendations and should not be used alone as a risk assessment (IUCN 2020; Kumschick et al. 2020, 2024). Impact assessments might be part of risk assessments, but other aspects (e.g. conservation purpose, feasibility and effectiveness of management, trade-offs among stakeholders, costs etc.) need to be considered before management recommendations can be given. Whether or not impacts from human-mediated introductions of alien populations are desirable or not needs to be decided by relevant stakeholders and is determined by multiple and sometimes conflicting value systems and ethical worldviews. EICAT might be useful to inform such decisions because it can elucidate in an objective, standardised and transparent way the known impacts of such introductions. We have now added some explanations (lines 514-518) to clarify these aspects.

All species are also consumed by other species, e.g. by detritivores. As a consequence, all species can also be beneficial and useful. What matters is to estimate how an anthropogenic change as introducing species cause on native biodiversity.

We wholeheartedly agree, and estimating how an anthropogenic change, such as the introduction of alien species, affects native biodiversity is exactly the focus of our analysis. However, we disagree that this necessarily involves a comparison between native and alien populations.

It's definitely true that some introduced herbivores can cause extinctions or massive range declines, which is especially likely on islands. The authors of the present manuscript did indeed find limited evidence of the local extirpation of a few species in 4 studies (of which several returned following the construction of exclosures). These of course are topics of conservation concern, as when native large herbivores cause similar impacts (e.g., the effects of endangered Key deer or reintroduced giraffe). However, these 4 studies hardly support the central title and claim of this manuscript.

We defined as strong negative impacts cases in which a native population has been locally or globally extirpated or declined because of introduced herbivores. Extirpation of 28 native species was found in 13 studies (and not in 4 studies). Additionally, 840 cases of population decline were detected in 148 studies. This information can be found in Supplementary Figure 1 and the source data file. We documented strong, far-reaching negative impacts and believe that both the title and central claim of the manuscript are supported by our results.

Instead, most of the negative evidence compiled in this article is exactly what you'd expect herbivores to do by definition of being herbivores: reducing the abundance and cover/biomass of their food resources (though this is actually difficult to parse from their data). Reducing abundance is not the same thing as causing extinctions. However, it should also be noted that local extirpations of herbivory sensitive species should be expected: many plants are restricted to spatial or temporal refugia due to native herbivores (e.g., succulent euphorbs in Africa (Cowling et al. 2010), the trees in Bond's giraffe study, etc).

As we outlined above, the relevant comparison here is how native species are impacted when an alien species is introduced compared to when it is not. Asking if the alien species has higher or lower impacts than in its native range or a "similar" native species is irrelevant in this context. Additionally, it would be difficult to define the native species against which the alien should be compared.

The authors' chief conclusion is to caution against reintroducing extinct large herbivores or proxies. However, the argument for rewilding large herbivores is to restore their top-down influences on ecosystems in regulating vegetation, especially dominant plant species (Svenning et al. 2019). This top-down trophic regulation was a part of Earth's history for ~55 million years until human-caused extinctions ~45,000-600 years ago (Svenning et al. 2024) and appears to play a strong role in biodiversity maintenance at broad scales, at reducing wildfire frequency and severity, at maintaining nutrient and seed dispersal, creating open habitats, and so on (Doughty et al. 2016, Malhi et al. 2016,

Karp et al. 2021). However, the way EICAT is designed, these same direct top-down influences are called 'negative' effects. As such, I do not see how the results of this paper have any bearing on recommendations for rewilding efforts. The real issue is a disagreement in values about what is 'good' or not in the world: something that science itself cannot assess ((Slobodkin 2001, Sagoff 2020).

We agree that many controversies about what conservationists deem as good or bad are rooted in different value systems and probably also differences in risk perception. Our results clearly show that the introduction of large mammalian herbivores has had many negative consequences for native species in current communities, but also some positive impacts. We conclude that the introduction of LMH species should not be assumed to have a clear positive effect on native species, and their risks should be carefully assessed in advance. If the overall impacts of introducing LMH species would be overwhelmingly positive, we are currently lacking the evidence for it, at least on native biodiversity. Proponents of rewilding programmes should convincingly document which positive impacts introduced LMHs have on native biodiversity, as average impacts (i.e. a balance of positive and negative impacts) or community metrics (e.g. diversity, richness) may overlook the detailed patterns of biodiversity alteration (lines 507-509).

In conservation decisions, it should also be clarified what is deemed as positive and negative. The examples above mix positive impacts for native species (analogous to EICAT+) with abiotic impacts (e.g. reduced fire frequency) or general statements about the disturbance status of an ecosystem (e.g. top-down trophic regulation), and it is unclear why these should be considered as positive. Since many ecosystems are shaped by fire, a general reduction in fire frequency might at some point be harmful for fire specialists. Similarly, if a reduction in the disturbance regime by fire is deemed as positive, it is not clear why an increased, large-scale top-down vegetation disturbance by foraging herbivores should be considered as positive; just because it existed before humans roamed the planet is a position difficult to defend. In contrast, EICAT clearly defines the entities (see also lines 68-71 and 542-54 now) that can profit or suffer (native species) and therefore how to assign positive and negative impacts. There are certainly other ways to classify impact direction; however, without clearly defined criteria, arguments in favour or against impacts caused by introduced species are prone to subjectivity.

Finally, 50% of introduced large herbivores are threatened or extinct in their native ranges (which the authors do cite). Eradications could cause the extinction of species in the wild. These animals are also known to be intelligent and cultural beings with strong emotional ties to each other and they had no choice but to be entangled in human conquest and empire building. Instead of calling them 'invaders' we could be equally problematic and anthropomorphic but more accurately so and call these animals 'escaped slaves'. Would we ask the same questions or treat their impacts similarly?

These aspects do not contradict our approach or results, but rather highlight alternative value perspectives that might inform broader conservation and management decisions but are not related to our paper. As stated before, EICAT and EICAT+ do not directly lead to management recommendations (e.g. eradication) and should not be used alone as a risk assessment (lines 514-518). However, the observation that many introduced large herbivores are threatened in their native range is relevant for their conservation and should be considered alongside their (predicted) impacts in their alien range. As a consequence, we have now added a new section regarding this aspect (466-484).

We have never advocated for any eradication or other lethal control measures against introduced herbivores in the manuscript. In fact, we acknowledge that such actions could cause suffering and raise ethical concerns due to the sentience of introduced herbivores. However, all negative impacts resulting from their introduction—such as competition for resources or disease transmission—inevitably involve a certain degree of suffering for the affected native species. This ethical dilemma has been for instance discussed by Latombe and co-authors (2025), a paper that we have also cited in our manuscript (see line 517). Since this aspect would far exceed the focus of our research, we will not further address the ethical dimensions of management strategies.

Lastly, we did not use the term “invaders” in our manuscript, and we refer to invasive species only in the introduction to indicate how they have been described in the literature.

I wish I had more constructive feedback to provide but I just do not see how the simplistic moral mathematics of EICAT, which lacks a proper control to make the claims it makes, can help us understand and respond appropriately to radical environmental change. I understand that EICAT and this type of reasoning is pervasive but that does not make it scientific. As far as I can tell, the authors’ results simply suggest that introduced large herbivores are, indeed, herbivores.

As such, I cannot support the publication of this article in as fine a journal as Nature Communications. If the authors had done something far more difficult, such as doing an EICAT analysis of the effects of native herbivores as well (the proper control to make any claim about the effects of introduced herbivores) then this article would be fascinating and a valuable contribution to the literature. I know that is a lot to ask though.

See previous responses.

We have the impression that some aspects of the EICAT(+) framework may have been misunderstood or misinterpreted, potentially leading to the perception that the methodology is unscientific, simplistic, or morally biased. To address these concerns, we have taken care to clarify and provide additional context in the revised manuscript regarding the points that may have been a source of misunderstanding. Specifically, we

have now clarified: why the impact of introduced species has been assessed by considering how they affected native species (lines 68-69); how we have distinguished between native and introduced species (lines 599-605); how we have minimized the presence of potential biases and how unlikely it is that they have affected our results (lines 302-334); why the comparison of impacts caused by introduced and native species might provide valuable scientific insights but addresses a different research question and thus has no influence on our concepts, methodology, findings and conclusions (lines 497-506).

In addition, we would like to stress that EICAT has been rigorously developed through an inclusive global consultation process led by the IUCN, incorporating feedback from hundreds of diverse stakeholders (Kumschick et al. 2024). The framework and its socio-economic derivative, SEICAT, have gained widespread acceptance, serving as critical tools in initiatives like the IPBES assessment on invasive alien species (Roy et al., 2023) and being formally recommended by the CBD at COP16 as mechanisms to support global biodiversity targets (<https://www.cbd.int/documents/CBD/COP/16/L.4>). Given this level of international endorsement and the robust scientific foundation underpinning EICAT, it is surprising and unfortunate to see it characterized in this manner. As it is unlikely that the global scientific community systematically applies unscientific or significantly biased tools, we believe that the methodology we followed, our overall results, and the clarifications provided in the revised manuscript and guided by the criticism of the reviewer, offer now a consistent, transparent and balanced approach that addresses pressing conservation challenges.

Forgione, L., Bacher, S., & Vimercati, G. (2022). Are species more harmful in their native, neonative or alien range? Insights from a global analysis of bark beetles. *Diversity and Distributions*, 28(9), 1832-1849.

Canavan, S., Richardson, D. M., Visser, V., Le Roux, J. J., Vorontsova, M. S., & Wilson, J. R. (2017). The global distribution of bamboos: assessing correlates of introduction and invasion. *AoB Plants*, 9(1), plw078.

Henry, A. K., & Sorte, C. J. (2022). Impact assessment of coastal marine range shifts to support proactive management. *Frontiers in Ecology and the Environment*, 20(3), 161-169.

IUCN. IUCN EICAT Categories and Criteria. The Environmental Impact Classification for Alien Taxa. IUCN EICAT Categories and Criteria: First Edition. (2020). doi:10.2305/IUCN.CH.2020.05.en.

Kumschick, S., Bacher, S., Bertolino, S., Blackburn, T. M., Evans, T., Roy, H. E., & Smith, K. (2020). Appropriate uses of EICAT protocol, data and classifications. *NeoBiota*, 193-212.

Kumschick, S., Bertolino, S., Blackburn, T. M., Brundu, G., Costello, K. E., De Groot, M., ... & Bacher, S. (2024). Using the IUCN Environmental Impact Classification for Alien Taxa to inform decision-making. *Conservation Biology*, 38(2), e14214.

Latombe G et al (2024) Ethical dilemma in conservation: a trolley problem thought experiment.

Roy, H. E. et al. IPBES Invasive Alien Species Assessment: Summary for Policymakers. IPBES (2023).

Afonso, B. C., L. M. Rosalino, J. Henriques, R. Tinoco Torres, J. Wauters, and J. Carvalho. 2023. The effects of wild ungulates on small mammals: a systematic review and meta-analysis. *Mammal review*.

Boltovskoy, D., N. M. Correa, L. E. Burlakova, A. Y. Karatayev, E. V. Thuesen, F. Sylvester, and E. M. Paolucci. 2021. Traits and impacts of introduced species: a quantitative review of meta-analyses. *Hydrobiologia* 848:2225–2258.

Bond, W. J., and D. Loffell. 2001. Introduction of giraffe changes acacia distribution in a South African savanna. *African journal of ecology* 39:286–294.

Coverdale, T. C., T. R. Kartzinel, K. L. Grabowski, R. K. Shriver, A. A. Hassan, J. R. Goheen, T. M. Palmer, and R. M. Pringle. 2016. Elephants in the understory: opposing direct and indirect effects of consumption and ecosystem engineering by megaherbivores. *Ecology* 97:3219–3230.

Cowling, R. M., A. Kamineth, M. Difford, and E. E. Campbell. 2010. Contemporary and historical impacts of megaherbivores on the population structure of tree euphorbias in South African subtropical thicket. *African journal of ecology* 48:135–145.

Daskin, J. H., and R. M. Pringle. 2016. Does primary productivity modulate the indirect effects of large herbivores? A global meta-analysis. *The Journal of animal ecology* 85:857–868.

Doughty, C. E., J. Roman, S. Faurby, A. Wolf, A. Haque, E. S. Bakker, Y. Malhi, J. B. Dunning Jr, and J.-C. Svenning. 2016. Global nutrient transport in a world of giants. *Proceedings of the National Academy of Sciences of the United States of America* 113:868–873.

Estes, J. A., and J. F. Palmisano. 1974. Sea otters: their role in structuring nearshore communities. *Science (New York, N.Y.)* 185:1058–1060.

Gordon, C. E., M. Greve, M. Henley, A. Bedetti, P. Allin, and J.-C. Svenning. 2023. Elephant rewilding affects landscape openness and fauna habitat across a 92-year period. *Ecological applications: a publication of the Ecological Society of America* 33:e2810.

Karp, A. T., J. T. Faith, J. R. Marlon, and A. C. Staver. 2021. Global response of fire activity to late Quaternary grazer extinctions. *Science* 374:1145–1148.

Lundgren, E. J., J. Bergman, J. Trepel, and E. le Roux. 2024. Functional traits—not nativeness—shape the effects of large mammalian herbivores on plant communities. *Science*.

Malhi, Y., C. E. Doughty, M. Galetti, F. A. Smith, J.-C. Svenning, and J. W. Terborgh. 2016. Megafauna and ecosystem function from the Pleistocene to the Anthropocene. *Proceedings of the National Academy of Sciences of the United States of America* 113:838–846.

Ripple, W. J., E. J. Larsen, R. A. Renkin, and D. W. Smith. 2001. Trophic cascades among wolves, elk and aspen on Yellowstone National Park's northern range. *Biological conservation* 102:227–234.

Sagoff, M. 2020. Fact and value in invasion biology. *Conservation biology: the journal of the Society for Conservation Biology* 34:581–588.

Slobodkin, L. 2001. The good, the bad and the reified. *Evol. Ecol. Res.* 3:91–105.

Svenning, J. C., R. T. Lemoine, and J. Bergman. 2024. The late-Quaternary megafauna extinctions: patterns, causes, ecological consequences, and implications for ecosystem management in the Anthropocene. *Cambridge Prisms* 2.

Svenning, J.-C., M. Munk, and A. Schweiger. 2019. Trophic rewilding: ecological restoration of top-down interactions to promote self-regulating biodiverse ecosystems. in N. Pettorelli, S. M. Durant, and J. T. du Toit, editors. *Rewilding*. Cambridge University Press.

Titcomb, G., B. F. Allan, T. Ainsworth, L. Henson, T. Hedlund, R. M. Pringle, T. M. Palmer, L. Njoroge, M. G. Campana, R. C. Fleischer, J. N. Mantas, and H. S. Young. 2017. Interacting effects of wildlife loss and climate on ticks and tick-borne disease. *Proceedings. Biological sciences* 284.

Barrett, P. Stiling, and Univ. of South Florida, Tampa, USA. 2006. Impacts of endangered Key deer herbivory on imperiled pine rocklands: a conservation dilemma? *Animal biodiversity and conservation* 29:165–178.

VanderMolen, M. S., and C. R. Webster. 2021. Influence of deer herbivory on regeneration dynamics and gap capture in experimental gaps, 18 years post-harvest. *Forest ecology and management* 501:119675.

Waldram, M. S., W. J. Bond, and W. D. Stock. 2008. Ecological engineering by a mega-grazer: White rhino impacts on a south African Savanna. *Ecosystems* 11:101–112.

Reviewer #2 (Remarks to the Author):

This is an interesting analysis of the impacts of invasive alien large herbivores. I think the paper could be a useful contribution to our understanding of the subject, but there are aspects of the analysis and discussion that need to be addressed.

Thank you for your positive comments on our analysis of the impacts of invasive alien large herbivores. We are pleased to hear that you find the paper a valuable contribution to the subject. We appreciate your constructive feedback and have carefully addressed the aspects of the analysis and discussion that needed further clarification or improvement. Please find detailed responses below.

Main comments

(1) Confirmation bias/controlling for effort. A significant concern I have about the analysis is the risk of confirmation bias in the underlying dataset. In the context of the global biodiversity, many scientists are undertaking research to identify negative impacts. This is perfectly valid because it is negative impacts that do harm to ecosystems. However, this means there may be more scientists looking/designing monitoring programs/experiments to identify negative impacts than positive impacts. In other words there is risk of confirmation bias when these data are used for comparative studies such as this. This means the dataset may contain differential effort. Indeed between lines 432 – 436, the authors point out the increase in interest in positive impacts of invasive species. This implicitly supports my concern that there has been less effort looking for positive impacts. The comparison of positive versus negative impacts is central to this manuscript. Please can the authors outline whether or not they controlled for this potential bias/differential effort? If not, I think they need to look at doing so. Further, I think this issue of potential bias in the underlying dataset should be explored in the Discussion section.

Thank you for raising the important concern of potential confirmation bias and differential effort in the underlying dataset. We fully acknowledge that much of the research on introduced species has historically focused on identifying negative impacts, which might indeed reflect an inherent bias in how data has been collected and reported. In particular, such a bias might lead us to expect finding more reports of negative than of positive impacts (which is what we found). However, the same outcome can be expected if there are indeed more negative than positive impacts; both mechanisms are difficult to disentangle. Since about a decade, positive impacts of alien species have received a lot of attention in the invasion community, thus there is now ample evidence and agreement that positive impacts exist and need to be considered. Several lines of argumentation which we present in the paper (lines 302-334) make us believe that the dataset we used does not exhibit a major systematic bias toward negative impacts: First, we have made efforts to include studies that assess both positive and negative outcomes of introduced species on native biodiversity, as demonstrated by the 407 observations related to positive impacts. Second, we found that the average magnitude of negative impacts is greater than that of positive impacts (Figure 3), which would be

unexpected if positive impacts were truly equal to or larger than negative impacts. Third, we would expect that large impacts (both negative and positive) are reported first, as they are more noticeable and easier to detect, with the reported magnitude declining over time. We found strong indications for this hypothesis for positive impacts, but only weak support for negative impacts, although there are more data points (Figure 4), indicating that there are not as many large positive impacts as there are large negative impacts. However, we agree that the risk of confirmation bias cannot be entirely ruled out with an opportunistically gathered dataset. To address this concern, we have now provided additional clarification in the manuscript regarding the steps taken to mitigate such biases in the dataset, including our data collection methods. We have also provided additional information (lines 315-321), conducted a novel statistical test (lines 319-320), and clarified the above reasoning to support our position that reporting bias was not a significant concern in our study. These changes can be found at lines 302-334.

(2) Defining nativeness. The authors don't really define what they mean by native/nativeness. I think this needs to be addressed and discussed. One example they use is of subspecies of *Cervus elaphus*. This is still the same species, and I am unconvinced that this is splitting the definition of nativeness too far for a broad-scale analysis like this. By definition as the same species, it is arguable whether different subspecies can be functionally considered "alien". Further, given the propensity for taxonomists to split and then lump sub-species, such definitions are fluid. Also, due to the fragmentation of populations, many geneticists consider it good practice to mix sub-populations to enhance genetic diversity and adaptive capacity; especially where inbreeding has occurred.

We thank the reviewer for the comment. We have now clarified the meaning of both terms "alien" and "native" at lines 599-605. We think that after our clarification, it is now more straightforward to understand why the consideration of subspecies such as those of *Cervus elaphus* is justified in our research. This said, we would like to stress that the number of impact observations associated with the two subspecies of *C. elaphus* is very limited (see Fig. 1) and has a negligible influence on the outcomes of the broader analysis.

Another consideration is the "refugee species concept" (see the work of Graham Kerley and colleagues). This is where species ranges have been drastically reduced over time, and extant populations have been confined to refuge habitat that may not be optimal. However, such sub-optimal habitat often become the "stereotype" for that species i.e. where it is considered "native". There are multiple issues with this for conservation, but a key one is that we may underestimate the former range of a species i.e. where we would consider it native. Under the criteria used in this manuscript, a species could be considered "alien" in places where it is not to have formerly occurred. Similarly, under climate change scenarios, the potential niche of a species may change, and it is legitimate to consider where a species might thrive outside our current definition of its

range. The authors recognise this, but it does raise the question about whether seeing “alienness” as a binary concept is too rigid.

We understand the point raised by the reviewer and feel that our definitions of “alien” and “native” (lines 599-605) now clarified this aspect. We agree that in some cases we may underestimate the “former range of a species i.e. where we would consider it native”. However, we do not think that this relates to the concept of nativeness (or non-nativeness) itself, but rather to our occasionally limited epistemological capacity to describe past and present geographical distribution and eco-evolutionary events (Essl et al. 2018). Our research is based on papers and reports in which the authors explicitly identified the impacting species as alien, thus relying on the authors’ judgement. Moreover, we made every effort to ensure that all impact observations were recorded within the alien range of the impacting species. This approach aligns with the IUCN EICAT categories and criteria, which clearly state: “EICAT is solely concerned with impacts in the alien range of a taxon, and data or observations from the native range should not be used in assessing impacts under EICAT”. Regarding species expanding due to climate change, on the contrary, they are bio-geographically distinct from introduced species (neonatives sensu Essl et al. 2018) and therefore they are not considered in our paper.

Another example of the complexities, is where a species has become extinct but it had important ecological functions for the ecosystem that it occupied. It is reasonable to consider niche substitutes as part of a rewilding program. Those species that are evolutionarily and geographically closer are probably less alien than those that are further away. I think alienness is therefore more of a continuum i.e. some species are more alien than others. I agree with the authors that caution should be exercised with such rewilding, but there is a risk that taking a binary view of impacts may miss opportunities for positive conservation outcomes. It would be good to see more of a discussion of these nuances around definitions of what an alien species is, and what the risks are.

As stated above and in line with commonly accepted definitions (e.g. Blackburn et al. 2011), we consider alienness (as nativeness) more like a binary concept (Essl et al. 2018), although our capacity to reconstruct under which circumstances humans have altered the biogeography of species can be limited in certain cases. However, we agree with the reviewer that rewilding should take into consideration how similar to now extinct taxa, both functionally and taxonomically, some introduced species might be in order to promote positive conservation opportunities. However, the similarity to native species doesn’t change the concept of “alienness”. We have expanded the section regarding rewilding at lines 485-497 now.

(3) Novel ecosystems and coexistence. While I recognise that the aim of this paper is to try and quantify the problem/impacts of native species, how does this apply to novel ecosystems? Many ecosystems are modified to a greater or lesser extent by humans

(this occurs on a continuum). Such modified ecosystems contain novel mixes of species. The focus of impact on native species while understandable, may miss the reality in many parts of the world. Many invasive species are very hard to control, and total eradication may be impossible. Therefore, managers in many situations have no choice but to manage for some form of uneasy coexistence between native and non-native ecosystem elements. I think it would be good to see more acknowledgement of this complexity and nuance when considering the implications of the findings of this analysis.

We recognize that nearly all ecosystems worldwide have experienced varying degrees of anthropogenic change, with some undergoing such extensive alterations that they qualify as novel ecosystems. These ecosystems are characterized by fundamental shifts in their species composition, structure, and ecological processes, diverging significantly and supposedly irreversibly from their pre-disturbance states. However, by focusing specifically on the impact of an alien species on a native species in a certain locality, we could more specifically identify how the latter species suffers or profits from the introduction of an alien species, thus minimizing the influence of other confounding factors (other alien species, habitat change etc...). This identification was therefore carried out whenever an impacting species was considered as native by the authors of the report, regardless of whether an ecosystem was regarded as more or less novel. In addition, we would like to emphasize that in over 2000 impact observations, we did not encounter any cases in which the authors faced difficulties identifying native species due to the system being so modified that it could be unequivocally classified as a novel ecosystem. That said, as we have pointed out before, EICAT(+) assessments do not directly lead to management recommendations like eradication, but rather provide an objective method for assessing ecological impacts. Management decisions should be based on a variety of factors, including knowledge on impact dynamics, feasibility and stakeholder values. We have now added some lines (514-518) to highlight this aspect.

Detailed comments.

Line 18 – comma after common and magnitude

Amended.

- See Main Comments regarding comparison of negative and positive effects

Line 72 – “population-level changes” – to native species?

Correct. We have amended this now at line 75.

Line 80 and 82 – see Main Comments. What is meant by “native” needs to be defined.

See general response on this aspect above.

Line 115 – “conservation perspective”. Please define what is meant by this?

We fully agree that “conservation perspective” was ambiguous. We have now re-written the sentence (line 68-69) into “ The EICAT(+) frameworks consider native biodiversity as the entity of conservation concern...”.

Line 154 – 155 - Please expand more i.e. given the importance of this to the analysis, I don't think it is enough to refer readers to another document.

We agree with the reviewer that more details are necessary to understand how the confidence has been assigned. We have now added a section (lines 584-587) to clarify this.

Line 166 - I suggest adding something like “that have established populations (see above” after species. The analysis is focussed on the 66 species that have established out of a total of 286. Please check whole manuscript and ensure clarity about where referring to established species.

Correct. We agree with the reviewer that it is important to highlight this here (line 289) and elsewhere (e.g. 469, 472, 478, 483, 599). Therefore, the text has now been amended.

Line 169-171 – See Main Comments. There is risk of bias/confirmation bias towards negative impacts in the dataset that needs to be addressed.

Line 189 – See Main Comments regarding risk of bias in dataset.

See above general response regarding biases and how we have now addressed this aspect in the revised manuscript (see lines 302-334).

Line 207 – Please provide examples of conspecifics.

We have now reported an example of conspecifics in the following sentence (lines 647-648).

Line 233 and 236 – “were” rather than “are” – please check tenses throughout manuscript.

Correct. Amended here (line 673) and elsewhere (lines 676-677).

Line 236 – “expected”. Amended.

Line 246 - See Main Comments regarding risk of bias in dataset.

Line 250 - See Main Comments regarding risk of bias in dataset.

Line 261 - See Main Comments regarding risk of bias in dataset.

Line 265 – 268 - See Main Comments regarding risk of bias in dataset.

Line 288 – 289 - See Main Comments regarding risk of bias in dataset.

Line 346 – 347 - See Main Comments regarding risk of bias in dataset.

Line 411 - See Main Comments regarding risk of bias in dataset. This needs to be discussed in the Discussion as a potential limitation.

Line 432 – 436 - See Main Comments regarding risk of bias in dataset. This reference to “recent” popularity implicitly implies that this may have been underreported/studied before this popularity.

See above general response regarding biases and how we have now addressed this aspect in the revised manuscript (see lines 302-334).

Line 482 – Wild boar are omnivores, so are quite distinct from the majority of large herbivores that are obligate herbivores. Given that, is it valid to refer to “most predation events” on islands in a general sense, when it refers to only an omnivore? Should wild boar be included in this analysis? To what extent is this omnivore influencing results in the rest of the analysis and dependent conclusions?

Thanks for pointing that out. In fact, the sentence was not properly phrased, as we meant that although some strong impacts were caused through predation, the majority of strong impacts were caused through other mechanisms. We have now restructured the whole paragraph, given additional details regarding the impacts on islands and added two new key references regarding the diet preferences of wild boars (see lines 377-390).

Line 493 – as above, omnivorous wild boar are lumped with obligate herbivores.

See at line 380 added references on the diet of wild boars. While this species is indeed omnivorous, it predominantly feeds on plant matter (see for instance in Schley and Roper 2003, where it is stated that “As regards volume, Janda (1958) and Massei, Genov & Staines (1996) both estimate that 86% of food consumed by the wild boar consists of vegetable matter, while Briedermann (1976), Abáigar (1993) and Fournier-Chambrillon et al. (1996) all give a figure of 95 or 96%.” Consequently, we believe that our choice to take into consideration wild boar in our analysis is pertinent.

Line 514 – 516 – What is the risk that invasive alien herbivores could help inflate populations of both native and invasive predators that could then enhance impact on native prey species through prey swapping? I think this needs to be discussed.

The hypothesis mentioned by the reviewer is certainly fascinating and reasonable, and we were indeed expecting to find evidence regarding such cases. Note that such indirect impacts would have been recorded through EICAT as caused via a mechanism called “indirect impacts through interactions with other species”. In spite of our expectation, we could not find cases reporting prey swapping. This may be related to the fact that the mechanism “provision of trophic resources,” which underpins the dynamic described by the reviewer, has only rarely been observed (see, for example, Fig. 2 and Supplementary

Table 1). Given that we could not substantiate this with evidence, we have preferred to not mention this aspect rather than risk unsupported speculation.

Line 566 – Please define rewilding and provide a reference. The following reference has the most recent definition:

Carver, S., Convery, I., Hawkins, S., Beyers, R., Eagle, A., Kun, Z., Van Maanen, E., Cao, Y., Fisher, M., Edwards, S.R. and Nelson, C., 2021. Guiding principles for rewilding. *Conservation Biology*, 35(6), pp.1882-1893.

However, if you are referring to “trophic rewilding” see:

Svenning, J.C., Pedersen, P.B., Donlan, C.J., Ejrnæs, R., Faurby, S., Galetti, M., Hansen, D.M., Sandel, B., Sandom, C.J., Terborgh, J.W. and Vera, F.W., 2016. Science for a wilder Anthropocene: Synthesis and future directions for trophic rewilding research. *Proceedings of the National Academy of Sciences*, 113(4), pp.898-906.

Many thanks for the comment. We have now defined both and referred to associated papers (line 488-493).

Essl, F., Bacher, S., Genovesi, P., Hulme, P. E., Jeschke, J. M., Katsanevakis, S., ... & Richardson, D. M. (2018). Which taxa are alien? Criteria, applications, and uncertainties. *BioScience*, 68(7), 496-509.

Blackburn, T. M., Pyšek, P., Bacher, S., Carlton, J. T., Duncan, R. P., Jarošík, V., ... & Richardson, D. M. (2011). A proposed unified framework for biological invasions. *Trends in ecology & evolution*, 26(7), 333-339.

Reviewer #3 (Remarks to the Author):

Comments on Bescond-Miochel et al.

In a novel quantitative synthesis, the authors have found that introductions of large mammalian herbivores outside their native range have “both harmed and benefited local native biodiversity, but negative consequences have largely surpassed positive outcomes, both in frequency and magnitude” and that such “negative impacts are more numerous, larger and often precede positive impact.” These results add new evidence to a debate that has unfolded on the opinion pages of journals and in the popular media, in which some researchers have argued that alien species impacts are biased toward negative effects. However, as the authors have correctly asserted (lines 52-53), there

has been no rigorous study demonstrating that the negative impacts of alien species on native biodiversity are biased. For example, there is abundant documented evidence of the major role of invasions as a driver of global extinctions, especially on islands worldwide, but no one has shown that these severe impacts are balanced out by positive effects.

Another noteworthy finding is that introduced mammalian herbivores that have had positive population-level impacts have done so mainly through indirect interactions with other species or disruptions to ecosystem properties that favor some certain taxa. I expect that this finding will add substance to the debate concerning the costs and benefits of alien megafaunal introductions – such as those at the center of proposed assisted colonization and rewilding schemes.

In summary, this study is a welcome and timely addition to the literature. I am confident that these findings will be widely cited, and probably debated, by ecologists and conservation biologists.

Thank you for your thoughtful and positive feedback. We're very pleased to know that the reviewer thinks that our findings will contribute to the ongoing debate on the impacts of alien species and are glad to see that our work may add substance to discussions on assisted colonization and rewilding schemes. Please find detailed responses to your specific comments below.

Specific comments:

Discussion: Are there cases where positive impacts clearly resulted in an enhanced ecosystem service or where negative impacts resulted in a disrupted or degraded ecosystem service? I understand that this may be outside the scope of the study.

Although in some papers we reviewed, the authors additionally showed or discussed enhancement or deterioration of ecosystem services following the introduction of large herbivores, we have decided to not address this aspect in our manuscript as it falls outside the primary scope of the study. Since ecosystem services refer to the benefits humans derive from ecosystems, but can vary or even compete depending on different values and interests across stakeholders, we have chosen to focus solely on ecological impacts on native species, as outlined in EICAT (and EICAT+). Please note that currently there is no consistent methodology to compare different ecosystem impacts according to their magnitude and direction (see also Bacher et al. 2023). However, at the end of our discussion, we have now clarified that EICAT(+) data “should not directly lead to management measures, but rather be used to inform local and national decision-making procedures on introduced LHM, alongside socio-economic considerations, ethical trade-offs and clarity in conservation goals” (lines 514-516).

Discussion: Can you discuss whether any of the species that caused strong negative impacts (at any location where they have been introduced) are imperiled in their native ranges? (e.g., Barbary sheep, mouflon, others?)

Many thanks for the suggestion. We have now extensively discussed this aspect and linked our research to a recent manuscript (Tedeschi et al. 2024, <https://doi.org/10.1111/conl.13069>) that explored how many threatened mammals established alien populations. The new section can be found at lines 466-484.

Discussion: You could mention that a potentially large set of indirect non-trophic effects have been ignored because such effects require very detailed study to recognize. However, we do know they exist; for example pigs create 'wallows' - muddy depressions that fill with water - and thus create habitat for mosquitoes (which could be classified as an indirect 'positive effect' on an insect), which in Hawaii likely exacerbated transmission of avian malaria (a clearly devastating effect on native avifauna).

Although such indirect mechanisms might indeed occur, we have restricted the impact data analysed to reported evidence, following the evidence-based approach of EICAT and EICAT+. As a consequence, we have aimed to focus exclusively on documented direct and indirect impacts of invasive alien LMH that are supported by empirical evidence, rather than supposed or potential but unconfirmed effects and mechanisms. This approach ensures that our conclusions remain robust and grounded in the available data, while acknowledging that undocumented impacts, such as the example of pigs creating mosquito habitats and facilitating avian malaria in Hawaii, may represent an underexplored aspect of their impacts.

Lines 64-67: The authors could mention that species threatened or endangered in their native ranges can nonetheless become invasive where they have been introduced, owing to context dependencies (the Conservation-Invasion Paradox).

We thank the reviewer for the great suggestion. We have now added multiple lines and some references regarding this aspect (lines 466-484).

Line 82: "Ecological dynamics" - I suggest using the term "ecoevolutionary dynamics" as it explicitly includes evolutionary context as a factor, and thus recognizes that disruption can arise for evolutionary mismatches (e.g., inadequate anti-herbivore defenses in native plants, lack of adaptation to disturbance from trampling, etc) and new interactions (e.g. facilitation of invasive plants).

Revised now in accordance with the reviewer's comment (line 86).

Line 90: Re: Introduced species can have positive effects if they "serve as an important novel food resource for native consumers" - This would only be a significant benefit to native consumers if food resources are otherwise limiting or of lower energetic value.

This is true, and was actually treated as such. Under EICAT+, simply consuming an alien species does not in itself count as a significant positive impact for a native species, if the native does not perform better than without the alien, i.e. on its normal foraging regime (Vimercati et al. 2022). Only if this mechanism leads to a measurable change in the performance of native individuals or above (e.g. increase in abundance), can we assign a magnitude. Paradoxically, if a native predator shifts its diet to include an alien prey—potentially taking advantage of the prey's naivete—but the alien prey has lower energetic value compared to native prey, this might be considered a negative impact. Such a scenario should be assessed using EICAT rather than EICAT+, as it reflects a harm rather than a benefit.

Line 93: "Impact magnitude would decline over time" – In this context, how did you treat local extinctions as a negative impact? The impact magnitude is obviously large but of limited duration and ends when the native population is wiped out, unless you classify it distinctly as a legacy effect.

Actually, we did not mean the magnitude of a specific case study, but the average impact magnitude reported across studies. To clarify this, we have now rephrased as follows: "Finally, we hypothesize that (3) due to their salience, negative and positive impacts of higher magnitude (strong impacts), such as local extinctions, have been identified first, and thus the reported impact magnitudes across studies would decline over time" (lines 96-99).

Lines 107-109: Perhaps emphasize here or elsewhere that several of the introduced species considered in this study might have significant positive or negative economic values that contrast with their ecological impacts; but those values are not being assessed here, otherwise species like hog deer (and others that pose a threat to crops) would have been ranked differently.

As mentioned above, we have chosen to focus solely on ecological impacts on native species, as outlined in EICAT (and EICAT+). However, at the end of our discussion, we have now clarified that EICAT(+) data "should not directly lead to management measures, but rather be used to inform local and national decision-making procedures on introduced LHM, alongside socio-economic considerations, ethical trade-offs and clarity in conservation goals" (lines 514-516).

Lines 153-155: The cited paper by Volery et al offers a vague methodology for assessing confidence: "When the assessor did not find evidence indicating that the true impact magnitude is likely to be the assigned one, a Low confidence level was assigned." This begs the question, why was an impact magnitude assigned if no evidence existed in the first place?

We agree with the reviewer that the paper by Volery et al. is ambiguous here and that further details are necessary to understand how the confidence has been assigned. To clarify Volery et al.: when it's "unclear if the assigned magnitude is likely to be the true magnitude" is not the same as "there is no evidence about its magnitude". For example, studies looking at gut content of alien herbivores can only confirm that a native plant species has been consumed (to the detriment of the plant, i.e. at least minor MN), but the true impact magnitude cannot be inferred from such studies. In these cases, the impact magnitude (at least MN) will be assigned a low confidence because there is no further evidence that this is the true magnitude. We have now added a section (lines 584-588) to clarify this aspect.

Lines 233-238: Can you offer an ecological reason for an observed decline in impact (e.g. native predators learning to capture the introduced herbivore)?

As stated above, we did not mean that the magnitude of a specific case study decreases but the magnitude reported yearly across studies. To clarify this, we have now rephrase as follows: Finally, we hypothesize that (3) due to their salience, negative and positive impacts of higher magnitude (strong impacts), such as local extinctions, have been identified first, and thus the reported impact magnitudes across studies would decline over time (lines 96-99).

Lines 258-261: "Species for which only positive impacts have been reported" include hog deer, Nilgai, red deer. Yet, according to the Global Invasive Species Database. negative ecological impacts of introduced red deer have been reported in South America (<https://www.iucngisd.org/gisd/species.php?sc=119>). Nilgai have been reported to damage mangroves in Texas (<https://scholarworks.utrgv.edu/etd/339/>). Perhaps I missed something, but does this imply that impacts are conservatively estimated in this study, or that some reports did not meet search criteria?

We thank the reviewer for pointing this out. Regarding red deer, we would like to emphasize that negative impacts of this species have been largely reported in our study. As shown in Figure 1, *Cervus elaphus* is the sixth species listed for negative impacts (104 in total). In the line mentioned by the reviewer we were specifically referring to two subspecies of red deer (which we listed separately; see explanation at lines 643-649), but reviewer's comment made us realize that mentioning these two subspecies was likely misleading here, as we referred to species. Therefore, we have now amended the sentence (see lines 121-127). And many thanks for making us aware of the report pertaining to Nilgai. While our extensive review aimed to capture the largest set of available literature, we cannot claim that it is complete, as some reports may not have been identified or were not available at the time of our review. We appreciate the effort made by the reviewer, and after screening the novel reference, we extracted two new impact observations (classified as MN and MN+) that have now been incorporated into the underlying dataset. Consequently, we reran all statistical analyses and re-designed the figures by using the updated dataset. This explains why results and tables, although

almost negligibly, differ now from the previous version of the manuscript. To reflect the new results, the sentence has also been revised (121-127).

Lines 462-463: "Similar ecosystem changes from woodlands to grasslands (including ferns) were promoted by widespread alien deer species such as *Cervus elaphus*". Although successional species (ferns) benefit from the introduced deer's activities, they do so at the expense of previously established species in the woodland community. Then why are no negative impacts assigned to *C. elaphus* in Figure 1?

Please see the response above.

Bacher S et al (2023) Chapter 4. Impacts of Invasive Alien Species on Nature, Nature's Contributionsto People, and Good Quality of Life. In: Thematic Assessment Report on Invasive Alien Species andTheir Control of the Intergovernmental Science-Policy Platform on Biodiversity and EcosystemServices. Roy, H. E., Pauchard, A., Stoett, P., and Renard Truong, T. (Eds.)

Tedeschi, L., Lenzner, B., Schertler, A., Biancolini, D., Essl, F. and Rondinini, C. (2024), Threatened Mammals With Alien Populations: Distribution, Causes, and Conservation. Conservation Letters. e13069. <https://doi.org/10.1111/conl.13069>

DETAILED RESPONSES TO THE REVIEWER'S COMMENTS

Please find our detailed responses in red below, with all changes highlighted using track changes in the marked-up copy.

Reviewer #2

The manuscript is much improved, and I thank the authors for the amount of work they have done to engage with and address the reviewers' comments. While I think the manuscript could be an important contribution, I continue to have a number of concerns that I will outline below. I note the line numbers did not correspond - so I found it difficult to verify every point made.

Thank you for your constructive and thoughtful feedback, as well as for recognizing the efforts made to improve the manuscript. We have carefully considered and addressed the concerns you raised, as detailed in our specific responses below. We also apologize for the previous mismatch in line numbers – as specified in our response letter, they referred to the marked-up copy rather than the revised manuscript. Please note that, in accordance with the journal guidelines, the line numbers now refer to the revised (clean) version of the manuscript to support a smoother review process.

(1) Bias. The authors have undertaken considerable work to address my concern about a bias towards studies focused on negative impacts. However, the changes they have made are essentially arguments of logic. If effects are compared, I think the bias/different effort needs to be controlled/accounted for in the modelling. This will give the reader more confidence in the comparisons.

We appreciate the reviewer's thoughtful suggestion to account for potential differences in sampling effort when comparing negative and positive impacts via the modelling. Conceptually, this is a reasonable concern. However, it is currently unclear how such differences in effort could be reliably evaluated. The mere fact that more studies report negative impacts is not sufficient to conclude that researchers preferentially or disproportionately study alien species expected to cause harms. It is entirely conceivable that negative impacts due to the introduction of alien species prevail. Considering our results, the underlying assumption behind a potential research bias is that scientists select their study systems based on expected outcomes prior to data collection (i.e., confirmation bias), and while this tendency could still apply to both positive and negative impacts, most researchers selected systems where negative impacts were suspected. If this were indeed the case, the observed outcomes of the studies (i.e., whether a negative or positive impact was found) could serve as a proxy for sampling effort aimed at detecting a particular type of impact. However, we find it unrealistic to assume that all studies on impacts were conducted by authors with a preconceived expectation (conscious or not) regarding the impact direction. It is unknown how many and which of the original studies were

conducted with an impact direction in mind. Thus, the bias in sampling effort for finding negative or positive impacts is unknown and we cannot include it in the analyses on impact numbers (as specified at lines 300-303).

Moreover, it is also not trivial to conceive a sampling design to collect new impact data that will account for all potential selection biases — for instance, which alien species should be chosen for an unbiased study, and which native species should be considered when measuring impacts? While the development of such unbiased study designs is important, it lies beyond the scope of the present paper, especially given that it is not clear which variables could be reliably included in the analysis to account for such biases, as suggested by the reviewer.

This said, we would like to highlight that in our revised manuscript, we have not only provided new logical arguments to address the reviewer's concern, but also used them to formulate and test several hypotheses, each accompanied by statistical analyses, in order to see whether there are **indications of relevant selection bias in numbers of impact studies** that might affect our conclusions; these are explained in detail in the Discussion section (lines 289-324):

1. The pattern of more negative than positive impacts is not restricted to few, well-studied species; it can be found in almost all species included in the study (e.g. see Fig 1). It also fits findings of the IPBES report, i.e. the most recent and comprehensive global synthesis study on impacts of invasive alien species: negative impacts outnumber positive impacts across taxa, regions, and realms (see Bacher et al. 2023, figure 4.2).
2. For reports that document both positive and negative impacts in LMH, i.e. reports whose study design and methods allowed the authors to capture changes in both directions, negative impacts still outnumbered positive impacts ($p < 0.001$).
3. Among reports that exclusively documented negative or positive impacts (231 vs. 21), the proportion of reports with a single observation was similar between the two groups (35% vs. 48%). This suggests that although some studies may have exhibited confirmation bias by selectively picking cases to highlight specific single impacts on native species, rather than conducting unbiased investigations across a broader range of species, this bias affected positive and negative impact studies similarly.
4. Temporal reporting of positive impacts was not increasing faster than that of negative impacts but remained stable over the last decades despite increased popularity of influential papers emphasizing the need to include positive impacts in assessments (some of these papers are mentioned in the manuscript).
5. If scientists were able to anticipate the likely direction of impacts, then one can assume that large impacts (positive and negative) would be preferentially chosen for studies and that it would be increasingly difficult over time to report new large impacts. However, the magnitude of reported positive impacts

declined faster over the years than that of negative impacts, indicating that strong positive impacts may have been incrementally more difficult to identify than negative ones, the opposite of what would be expected if a large selection bias towards negative impacts would exist.

More generally, the statistical models that we applied to identify factors explaining variation in impact magnitude (GLMM fitted by maximum likelihood) produce unbiased parameter estimates and standard errors even with unbalanced data (Zuur 2009; Bolker et al. 2009; Schielzeth et al. 2020). Thus, the comparison of impact magnitudes and directions, and identification of factors associated with magnitude should not be affected by unequal numbers of negative and positive impacts.

Lastly, other potential sources of biases might have affected our results, such as the uncertainty associated with the refugee species concept advanced by the reviewer. If some populations that we considered as alien were introduced in locations unknowingly formerly inhabited by the species (i.e. formerly native) and these populations caused proportionally more negative and severe impacts than "true" alien populations, this might have distorted our results (as acknowledged now at lines 333-344). However, accounting for this in our model did not improve model fit factor (see below response to reviewer comment number 3), confirming the robustness of our results.

(2) Nativeness - *Cervus elaphus*. I remain unconvinced by splitting impact to the sub-species level, nor that sub-species are necessarily alien. Many populations/ecologically significant units are an artifact of fragmentation of former connected clines. The classification of sub-species is constantly being reviewed. I think this distinction is not necessary in the context of the subject of impact. Also, I think managers would be alarmed if they have to think about sub-species x impact given all the factors they already have to manage at once.

We understand the reviewer's concerns regarding the relevance and reliability of distinguishing impacts at the subspecies level, particularly considering ongoing taxonomic revisions and the complexity this may introduce for managers. Based on the reviewer's argument but also the taxonomic uncertainty related to two sub-species of red deer included initially in our analysis, we removed the two impact data points from the dataset and repeated all analyses, updated tables and redrew all figures. The results showed minimal changes (as the reviewer can see in the marked-up version of the MS for comparison), and our overall conclusions were unaffected.

That said, we would like to clarify our initial rationale for including subspecies-level distinctions and further illustrate to the reviewer why the EICAT (+) approach can be extended to sub-species and other infraspecific units. When subspecies introductions are well-documented and ecologically distinct from native lineages, the resulting impacts may differ meaningfully from those caused by native populations. We would also like to highlight that such subspecies—introduced to areas where they did not

previously occur and thus are potentially considered alien—can have significant impacts on wild populations of the same species, particularly when those native populations are of high conservation concern and already recognized as such by managers. One example is the impact of domesticated forms of Atlantic salmon used in aquaculture on wild populations of Atlantic salmon through transmission of fish lice and other diseases, which multiply in breeding sites due to the artificially high fish abundance (Ford and Myers 2008, Gargan et al 2012). Since the domesticated form differs significantly, both genetically and phenotypically, from wild populations (Gross 1998), we believe that the concept of impacts caused by alien subspecies—or domesticated forms—on wild populations of the same species is analogous to the impacts caused by alien species on native populations. As such, these impacts can be of genuine conservation concern and should be considered by practitioners when making management decisions.

(3) Refugee species concept (RSC) and nativeness. I don't think the authors fully recognise the potential impact of the RSC on their binary categorisation of native versus alien. A key point about RSC is that most ecologists and conservation biologists etc don't currently consider its effect and impacts on species distribution and habitat stereotypes. Thus 'relying on the authors' judgement' reinforces this. I reiterate that I think the implications of RSC on categorizing alien or native is important.

We thank the reviewer for further clarifying their concerns regarding the implications of the Refugee Species Concept (RSC) for our binary classification of species as native or alien. The Refugee Species Concept (RSC) postulates that some extant populations cannot any more access their optimal habitat (e.g. due to human limitation, management) and are confined to suboptimal areas (Kerley et al. 2012). The authors of this concept illustrate it with European bison, which they hypothesize might not be a forest species (as it is now and was in the known past), but rather a grassland species, but humans confined it to and kept it in forests. More generally, the RSC hypothesizes that some species in their current distribution are not in their optimal (native) habitat and when they are introduced in other habitat types might be better adapted, perform better, and, hypothetically, might have larger (or more neutral “different”) impacts. While this is an interesting concept, the evidence that many species are concerned seems to be rather limited.

Our study could be impacted by the RSC if some species introductions took place in locations that were unknowingly formerly inhabited by the species, and the impacts of these “formerly native” populations differ in magnitude and directions from those of “truly alien” population. One key problem is to identify such locations, as they are unknown by definition. We hypothesized that populations located close to the currently known native range, i.e. on the same continent or in locations directly adjacent to the native area even on a different continent (e.g. North Africa -> Spain; Papua New Guinea -> Indonesia), are more likely to be “formerly native” than populations located far away, i.e. on different continents. However, a factor

discriminating these two types of introduction locations in the GLMM analyses did not improve model fit and was not considered in model selection procedures. Thus, we didn't find indications that the RSC influenced our findings on impact magnitude and direction. That said, we thank the reviewer for their suggestion and have addressed this aspect in our discussion, offering recommendations on how it could be incorporated into the analyses, along with additional references to explain the RSC concept.

This is explained in detail in a fully novel section at lines 325-344.

Reviewer #3

I have read the manuscript and the authors' responses, and I found them to have addressed my concerns quite effectively. I am impressed by how the authors' handled the reviewers' comments in general.

Thank you for your kind feedback and for carefully reviewing the revised manuscript. We are pleased to hear that our responses addressed your concerns.

Literature cited

Bolker, B. M., Brooks, M. E., Clark, C. J., Geange, S. W., Poulsen, J. R., Stevens, M. H. H., & White, J.-S.-S. (2009). Generalized linear mixed models: A practical guide for ecology and evolution. *Trends in Ecology & Evolution*, 24, 127–135.

<https://doi.org/10.1016/j.tree.2008.10.008>

Ford, J. S., & Myers, R. A. (2008). A global assessment of salmon aquaculture impacts on wild salmonids. *PLoS biology*, 6(2), e33.

Gargan, P. G., Forde, G., Hazon, N., Russell, D. J. F., & Todd, C. D. (2012). Evidence for sea lice-induced marine mortality of Atlantic salmon (*Salmo salar*) in western Ireland from experimental releases of ranched smolts treated with emamectin benzoate. *Canadian Journal of Fisheries and Aquatic Sciences*, 69(2), 343-353.

Gross, M. R. (1998). One species with two biologies: Atlantic salmon (*Salmo salar*) in the wild and in aquaculture. *Canadian Journal of Fisheries and Aquatic Sciences*, 55(S1), 131-144.

Kerley, G. I., Kowalczyk, R., & Cromsigt, J. P. (2012). Conservation implications of the refugee species concept and the European bison: king of the forest or refugee in a marginal habitat?. *Ecography*, 35(6), 519-529.

Schielzeth, H., Dingemanse, N. J., Nakagawa, S., Westneat, D. F., Alaguela, H., Teplitsky, C., ... & Araya-Ajoy, Y. G. (2020). Robustness of linear mixed-effects models to violations of distributional assumptions. *Methods in ecology and evolution*, 11(9), 1141-1152.

Zuur, A. F., Ieno, E. N., Walker, N. J., Saveliev, A. A., & Smith, G. M. (2009). *Mixed effects models and extensions in ecology with R*. London, UK: Springer.

DETAILED RESPONSES TO THE REVIEWER'S COMMENTS

Please find our detailed responses in red below

Reviewer #2 (Remarks to the Author):

I thank the authors for the extensive amount of work they have undertaken in response to my comments. On the matters I raised, my key suggestions were to encourage the authors to discuss the caveats of their analysis. They have now done this very well, and also done additional analyses. This is excellent.

We are pleased that the reviewer appreciated our efforts in addressing their constructive feedback. We would like to take this opportunity to thank them once again for encouraging us to more thoroughly discuss the caveats of our analysis and to conduct additional tests.

I thank the authors for resolving my point about *Cervus elaphus* sub-species. At the same time, I do take their excellent point about domestic and wild forms of Atlantic salmon. They may wish to add some words on scenarios where sub-species could be damaging if they wish to make a point about it - but I don't mind i.e. not essential from my perspective.

We are also pleased the reviewer found convincing our point regarding domestic and wild forms, and more generally the impact between infraspecies units. Since we followed the reviewer's feedback and removed from the analysis the two data points concerning the impacts of a *Cervus elaphus* sub-species on other sub-species, we do not consider it necessary to further elaborate on this aspect in the manuscript. This issue is also very marginal in comparison to the broader patterns and conclusions presented in our study, as previously acknowledged by the reviewer.

I thank the authors for the extra analysis regarding refugee species concept. This demonstrates the robustness and rigour of their work. I do slightly disagree that evidence is limited for RSC - for example see:

Smith, K.J., Pierson, J.C., Evans, M.J., Gordon, I.J. and Manning, A.D., 2024. Continental-scale identification and prioritisation of potential refugee species; a case study for rodents in Australia. Ecography, 2024(9), p.e07035.

Regarding the refugee species concept, we are happy that the reviewer found our extra-analysis useful and instrumental in testing the robustness of our work. We also agree with the reviewer that the paper they mentioned, and we cited (Smith et al. 2024) highlights that this phenomenon may be more common than previously thought. However, relatively few studies have investigated this in recent years. We hope that our additional analysis and discussion will help raise attention toward this aspect and encourage further research—particularly within the field of biological invasions.